# DINO-WM: World Models on Pre-trained Visual Features enable Zero-shot Planning

## Abstract

The ability to predict future outcomes given control actions is fundamental for physical reasoning. However, such predictive models, often called world models, have proven challenging to learn and are typically developed for task-specific solutions with online policy learning. We argue that the true potential of world models lies in their ability to reason and plan across diverse problems using only passive data, without requiring online interactions with the environment. Concretely, we require world models to have the following three properties: 1) be trainable on offline, pre-collected trajectories, 2) support test-time behavior optimization, and 3) facilitate task-agnostic reasoning. To realize this, we present DINO World Model (DINO-WM), a new method to model visual dynamics without reconstructing the visual world. DINO-WM leverages spatial patch features pre-trained with DINOv2, enabling it to learn from offline behavioral trajectories by predicting future patch features. This design allows DINO-WM to achieve observational goals through action sequence optimization, facilitating task-agnostic behavior planning by treating desired goal patch features as prediction targets. We evaluate DINO-WM across various domains, including maze navigation, tabletop pushing, and particle manipulation. Our experiments demonstrate that DINO-WM can generate zero-shot behavioral solutions at test time without relying on expert demonstrations, reward modeling, or pre-learned inverse models. Notably, DINO-WM exhibits strong generalization capabilities compared to prior state-of-the-art work, adapting to diverse task families such as arbitrarily configured mazes, push manipulation with varied object shapes, and multi-particle scenarios.

## 1 Introduction

Robotics and embodied AI have seen tremendous progress in recent years. Advances in imitation learning and reinforcement learning have enabled agents to learn complex behaviors across diverse tasks (Lee et al., 2024; Zhao et al., 2023; Ma et al., 2024; Hafner et al., 2024; Hansen et al., 2024; Agarwal et al., 2022; Haldar et al., 2024) (Jia et al., 2024). Despite this progress, generalization remains a major challenge (Zhou et al., 2023). Existing approaches predominantly rely on policies that, once trained, operate in a feed-forward manner during deployment—mapping observations to actions without any further optimization or reasoning. Under this framework, successful generalization inherently requires agents to possess solutions to all possible tasks and scenarios once training is complete, which is only possible if the agent has seen similar scenarios during training (Brohan et al., 2023b;a; Reed et al., 2022; Etukuru et al., 2024). However, it is neither feasible nor efficient to learn solutions for all potential tasks and environments in advance.

Instead of learning the solutions to all possible tasks during training, an alternate is to fit a dynamics model on training data and optimize task-specific behavior during runtime. These dynamics models, also called world models (Ha & Schmidhuber, 2018), have a long history in robotics and control (Sutton, 1991; Todorov & Li, 2005; Williams et al., 2017). More recently, several works have shown that world models can be trained on raw observational data (Hafner et al., 2019; 2024; Micheli et al., 2023; Robine et al., 2023; Hansen et al., 2024). This enables flexible use of model-based optimization to obtain policies as it circumvents the need for explicit state-estimation. Despite this, significant challenges still remain in its use for solving general-purpose tasks.

To understand the challenges in world modeling, let us consider the two broad paradigms in learning world models: online and offline. In the online setting, access to the environment is often required

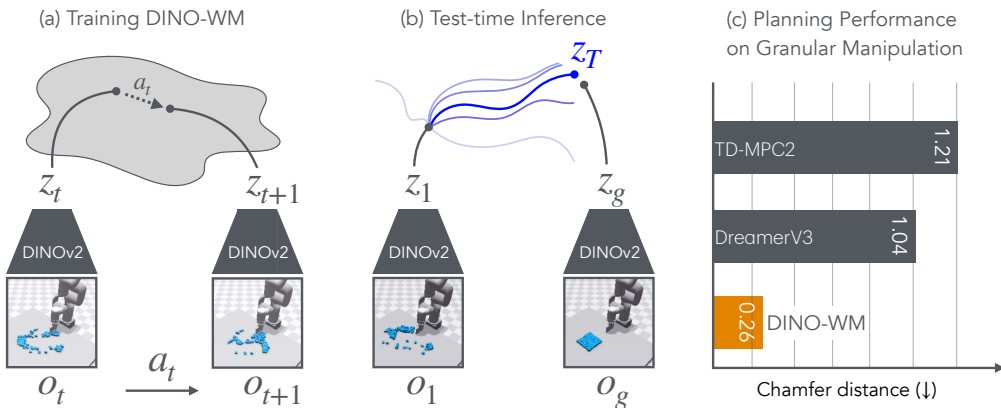

Figure 1: We present DINO-WM, a method for training visual models by using pretrained DINOv2 embeddings of image frames (a). Once trained, given a target observation $o_T$, we can directly optimize agent behavior by planning through DINO-WM using model predictive control (b). The use of pretrained embeddings significantly improves performance over prior state-of-the-art world models (c).

so data can be continuously collected to improve the world model, which in turn improves the policy and the subsequent data collection. However, the online world model is only accurate in the cover of the policy that was being optimized. Hence, while it can be used to train powerful task-specific policies, it requires retraining for every new task even in the same environment. Instead, in the offline setting, the world model is trained on an offline dataset of collected trajectories in the environment, which removes its dependence on the task specificity given sufficient coverage in the dataset. However, when required to solve a task, methods in this domain require strong auxiliary information to overcome the lack of dense coverage on the task-specific domain. This auxiliary information can take the form of expert demonstrations (Pathak et al., 2018), structured keypoints (Ko et al., 2023; Wen et al., 2024), access to pretrained inverse models (Du et al., 2023; Ko et al., 2023) or dense reward functions (Ding et al., 2024), all of which reduce the generality of using offline world models. The central question to building better offline world models is if there is alternate auxiliary information that does not compromise its generality?

In this work, we present DINO-WM, a new and simple method to build task-agnostic world models from an offline dataset of trajectories. DINO-WM models the world dynamics on compact embeddings of the world, rather than the raw observations themselves. For the embedding, we use pretrained patch-features from the DINOv2 model, which provides both a spatial and object-centric representation prior. We conjecture that this pretrained representation enables robust and consistent world modeling, which relaxes the necessity for task-specific data coverage. Given these visual embeddings and actions, DINO-WM uses the ViT architecture to predict future embeddings. Once this model is trained on the offline dataset, planning to solve tasks is constructed as visual goal reaching, i.e. to reach a future desired goal given the current observation. Since the predictions by DINO-WM are high quality (see Figure 4), we can simply use model predictive control with inference-time optimization to reach desired goals without any extra information during testing.

DINO-WM is experimentally evaluated on four environment suites spanning maze navigation, sliding manipulation, and particle manipulation tasks. Our experiments reveal the following findings:

- DINO-WM produce high-quality future world modeling that can be measured by improved visual reconstruction from trained decoders. On LPIPS metrics for our hardest tasks, this improves upon prior state-of-the-art work such as RSSM by 56% (See Section 4.7).

- Given the latent world models trained using DINO-WM, we show high success for reaching arbitrary goals on our hardest tasks, improving upon prior work by 83% (See Section 4.3).

- DINO-WM can be trained across environment variations within a task family (e.g. different maze layouts for navigation or different object shapes for manipulation) and achieve higher rates of success compared to prior work (See Section 4.5).

Code and models for DINO-WM will be open-sourced to ensure reproducibility and videos of policies are made available on our project website: https://anon-dino-wm.github.io.

## 2 RELATED WORK

We build on top of several works in building world models, optimizing them, and using compact visual representations. For conciseness, we only discuss the ones most relevant to DINO-WM.

**Model-based Learning:** Learning from models of dynamics has a rich literature spanning the fields of control, planning, and robotics (Sutton, 1991; Todorov & Li, 2005; Astolfi et al., 2008; Holkar & Waghmare, 2010; Williams et al., 2017). Recent works have shown that modeling dynamics and predicting future states can significantly enhance vision-based learning for embodied agents across various applications, including online reinforcement learning (Hafner et al., 2024; Micheli et al., 2023; Hansen et al., 2024; Robine et al., 2023), exploration (Mendonca et al., 2021; 2023a; Sekar et al., 2020), planning (Watter et al., 2015) (Finn & Levine, 2017; Ebert et al., 2018; Hafner et al., 2019), and imitation learning (Pathak et al., 2018). Several of these approaches initially focused on state-space dynamics (Deisenroth & Rasmussen, 2011; Chua et al., 2018; Lenz et al., 2015; Nagabandi et al., 2019), and have since been extended to handle image-based inputs, which we address in this work. These world models can predict future states in either pixel space (Finn & Levine, 2017; Ebert et al., 2018; Ko et al., 2023; Du et al., 2023) or latent representation space (Yan et al., 2021). Predicting in pixel space, however, is computationally expensive due to the need for image reconstruction and the overhead of using diffusion models (Ko et al., 2023). On the other hand, latent-space prediction is typically tied to objectives of reconstructing images (Hafner et al., 2019; 2024; Micheli et al., 2023), which raises concerns about whether the learned features contain sufficient information about the task. Moreover, many of these models incorporate reward prediction (Hafner et al., 2024; Micheli et al., 2023; Robine et al., 2023), or use reward prediction as auxiliary objective to learn the latent representation (Hansen et al., 2024; 2022), inherently making the world model task-specific. In this work, we aim to decouple task-dependent information from latent-space prediction, striving to develop a versatile and task-agnostic world model capable of generalizing across different scenarios.

**Generative Models as World Models:** With the recent excitement of large scale foundation models, there have been initiatives on building large-scale video generation world models conditioned on agent's actions in the domain of self-driving (Hu et al., 2023), control (Yang et al., 2023; Bruce et al., 2024), and general-purpose video generation (Liu et al., 2024). These models aim to generate video predictions conditioned on text or high-level action sequences. While these models have demonstrated utility in downstream tasks like data augmentations, their reliance on language conditioning limits their application when precise visually indicative goals need to be reached. Additionally, the use of diffusion models for video generation makes them computationally expensive, further restricting their applicability for test-time optimization techniques such as MPC. In this work, we aim to build a world model in latent space rather than in the raw pixel space, which enables more precise planning and control.

**Pretrained Visual Representations:** Significant advancements have been made in the field of visual representation learning, where compact features that capture spatial and semantic information can be readily used for downstream tasks. Pre-trained models like ImageNet pre-trained ResNet (He et al., 2016), I-JEPA (Assran et al., 2023), and DINO (Caron et al., 2021; Oquab et al., 2024) for images, as well as V-JEPA (Bardes et al., 2024) for videos, and R3M (Nair et al., 2022), MVP (Xiao et al., 2022) for robotics have allowed fast adaptation to downstream tasks as they contain rich spatial and semantic information. While many of these models represent images using a single global feature, the introduction of Vision Transformers (ViTs) (Dosovitskiy et al., 2021) has enabled the use of pre-trained patch features, as demonstrated by DINO (Caron et al., 2021; Oquab et al., 2024). DINO employs a self-distillation loss that allows the model to learn representations effectively, capturing semantic layouts and improving spatial understanding within images. In this work, we leverage DINOv2's patch embeddings to train our world model, and demonstrate that it serves as a versatile encoder capable of handling multiple precise tasks.

## 3 DINO WORLD MODELS

**Overview and Problem formulation:** Our work follows the vision-based control task framework, which models the environment as a partially observable Markov decision process (POMDP). The POMDP is defined by the tuple $(\mathcal{O}, \mathcal{A}, p)$, where $\mathcal{O}$ represents the observation space, and $\mathcal{A}$ denotes

Figure 2: Architecture of DINO-WM. Given observations $o_{t-k:t}$, we optimize the sequence of actions $a_{t:T-1}$ to minimize the predicted loss to the desired goal $o_g$. All forward computation is done in the latent space $z$. Here $p_\theta$ indicates DINO-WM's dynamics model, which is used for making future predictions.

the action space. The environment's dynamics are modeled by the transition distribution $p(o_{t+1} \mid o_{\leq t}, a_{\leq t})$, which predicts future observations based on past actions and observations.

In this work, we aim to learn task-agostnic world models from pre-collected offline datasets, and use these world models to perform visual reasoning and control at test time. At test time, our system starts from an arbitrary environment state and is provided with a goal observation in the form of an RGB image, in line with prior works (Wu et al., 2020; Ebert et al., 2018; Mendonca et al., 2023b), and is asked to perform a sequence of actions $a_0, ..., a_T$ such that the goal state can be achieved. This approach differs from world models used in online reinforcement learning (RL) where the objective is to optimize rewards for a fixed set of tasks at hand (Hafner et al., 2024; Hansen et al., 2024), or from text-conditioned world models, where goals are specified through text prompts (Du et al., 2023; Ko et al., 2023).

## 3.1 DINO-BASED WORLD MODELS (DINO-WM)

We model the dynamics of the environment in the latent space. More specifically, at each time step $t$, our world model consists of the following components:

$$
\begin{aligned}
\text{Observation model:} \quad & z_t \sim \text{enc}_\theta(z_t \mid o_t) \\
\text{Transition model:} \quad & z_{t+1} \sim p_\theta(z_{t+1} \mid z_{t-H:t}, a_{t-H:t}) \\
\text{Decoder model (optional for visualization):} \quad & \hat{o}_t \sim q_\theta(o_t \mid z_t)
\end{aligned}
$$

where the observation model encodes image observations to latent states $z_t$, and the transition model takes in a history of past latent states of length $H$. The decoder model takes in a latent $z_t$, and reconstruct the image observation $o_t$. We use $\theta$ to denote the parameters of these models. Note that our decoder is entirely optional, as the training objectives for the decoder is independent for training the rest part of the world model. This eliminates the need to reconstruct images both during training and testing, which reduces computational costs compared to otherwise coupling together the training of the observational model and the decoder, as in (Hafner et al., 2024; Micheli et al., 2023).

DINO-WM models only the information available from offline trajectory data in an environment, in contrast to recent online RL world models that also require task-relevant information, such as rewards (Hansen et al., 2022; 2024; Hafner et al., 2020), discount factors (Hafner et al., 2022; Robine et al., 2023), and termination conditions (Hafner et al., 2024; Micheli et al., 2023).

### 3.1.1 OBSERVATION MODEL

With the goal of learning a generic world model across many environments and the real world, we argue that the observation model should 1) be task and environment independent, and 2) contain rich spatial information which is crucial in navigation and manipulation tasks. Contrary to previous works where the observation model is always learned for the task at hand (Hafner et al., 2024), we argue instead that it is not always possible for world models to learn an observation model from scratch when facing a new environment, as perception is a general task that can be learned from the large corpus of internet data. Therefore, we choose the out-of-the-box pre-trained DINOv2

model as our world model's observation model, as it has been shown to excel at object detection, semantic segmentation, and depth estimation tasks which require substantial spatial understanding. The observation model is kept frozen throughout both training and testing time. At each time step $t$, it encodes an image $o_t$ to patch embeddings $z_t \in \mathbb{R}^{N \times E}$, where $N$ denotes the number of patches, and $E$ denotes the embedding dimension. This process is visualized in Figure 2.

### 3.1.2 Transition Model

We adopt the ViT architecture (Dosovitskiy et al., 2021) for the transition model due to its suitability for processing patch features. We modify the model by removing the tokenization layer, as it operates on patch embeddings, effectively transforming it into a decoder-only transformer. We further make a few modifications to the architecture to allow for additional conditioning on proprioception and controller actions.

Our transition model takes in a history of past latent states $z_{t-H:t-1}$ and actions $a_{t-H:t-1}$, where $H$ is a hyperparameter denoting the context length of the model, and predicts the latent state at next time step $z_t$. To properly capture the temporal dependencies, where the world state at time $t$ should only depend on previous observations and actions, we implement a causal attention mechanism in the ViT model, enabling the model to predict latents autoregressively at a frame level. Specifically, each patch vector $z_t^i$ for the latent state $z_t$ attends to $\{z_{t-H:t-1}^i\}_{i=1}^N$. This is different from past work (Micheli et al., 2023) which similarly represents each observation as a sequence of vectors, but autoregressively predicts $z_t^i$ at a token level, attending to $\{z_{t-H:t-1}^i\}_{i=1}^N$ as well as $\{z_t^i\}_{i=1}^{<k}$. We argue that predicting at a frame level and treating patch vectors of one observation as a whole better captures global structure and temporal dynamics, modeling dependencies across the entire observation rather than isolated tokens, leading to improved temporal generalization.

To model the effect of the agent's action to the environment, we condition the world model's predictions on these actions. Specifically, we concatenate the $K$-dimensional action vector, mapped from the original action representation using a multi-layer perceptron (MLP), to each patch vector $z_t^i$ for $i = 1, \ldots, N$. When proprioceptive information is available, we incorporate it similarly by concatenating it to the observation latents, thereby integrating it into the latent states.

We train the world model with teacher forcing. During training, we slice the trajectories in to segments of length $H + 1$, and compute a latent consistency loss on each of the $H$ predicted frames. For each frame, we compute

$$\mathcal{L}_{pred} = \|p_\theta\left(\text{enc}_\theta(o_{t-H:t}), \phi(a_{t-H:t})\right) - \text{enc}_\theta\left(o_{t+1}\right)\|^2 \tag{1}$$

where $\phi$ is the action encoder model that can map actions to higher dimensions. Note that our world model training is entirely performed in latent space, without the need to reconstruct the original pixel images.

### 3.1.3 Decoder for Interpretability

To aid in visualization and interpretability, we use a stack of transposed convolution layers to decode the patch representations back to image pixels, similar as in (Razavi et al., 2019). Given a pre-collected dataset, we optimize the parameters $\theta$ of the decoder $q_\theta$ with a simple reconstruction loss defined as:

$$\mathcal{L}_{rec} = \|q_\theta(z_t) - o_t\|^2, \quad \text{where} \quad z_t = \text{enc}_\theta(o_t) \tag{2}$$

The training of the decoder is entirely independent of the transition model training, offering several advantages: **1)** The quality of the decoder does not affect the world model's reasoning and planning capabilities for solving downstream tasks, and **2)** During planning, there is no need to reconstruct raw pixel images, thereby reducing computational costs. Nevertheless, the decoder remains valuable as it enhances the interpretability of the world model's predictions.

### 3.2 Visual Planning with DINO-WM

Arguably, to evaluate the quality of the world model, it needs to be able to allow for downstream reasoning and planning. A standard evaluation metric is to perform trajectory optimization at test time with these world models and measure the performance. While the planning methods themselves are fairly standard, it serves as a means to emphasize the quality of the world models. For this purpose,

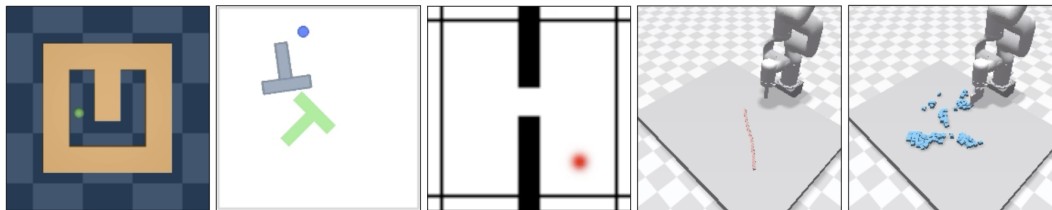

Figure 3: We evaluate DINO-WM on 5 environment suites, from left to right: PointMaze, Push-T, Two Room, Rope Manipulation, and Granular Manipulation.

our world model receives the current observation $o_0$ and a goal observation $o_g$, both represented as RGB images. We formulate planning as the process of searching for a sequence of actions that the agent would take to reach $o_g$. To achieve this, we employ model predictive control (MPC), which facilitates planning by considering the outcomes of future actions.

We utilize the cross-entropy method (CEM), a stochastic optimization algorithm, to optimize the sequence of actions at each iteration. The planning cost is defined as the mean squared error (MSE) between the current latent state and the goal's latent state, given by

$$\mathcal{C} = \|\hat{z}_T - z_g\|^2, \quad \text{where} \quad \hat{z}_t = p(\hat{z}_{t-1}, a_{t-1}), \quad \hat{z}_0 = \text{enc}(o_0), \quad z_g = \text{enc}(o_g). \quad (3)$$

The MPC framework and CEM optimization procedure are detailed in Appendix A.6.1. Since our world model is differentiable, a possibly more efficient approach is to optimize this objective through gradient descent (GD), allowing the world model to directly guide the agent toward a specific goal. The details of GD are provided in Appendix A.6.2. However, we empirically observe that CEM outperforms GD in our experiments. We hypothesize this is due to our choice to not constrain the terrain smoothness of the world model during training, potentially leading to issues with the gradient. Full results for both planners can be found in Appendix A.6.3.

## 4 Experiments

Our experiments are designed to address the following key questions: **1)** Can we effectively train DINO-WM using pre-collected offline datasets? **2)** Once trained, can DINO-WM be used for visual planning? **3)** To what extent does the quality of the world model depend on pre-trained visual representations? **4)** Does DINO-WM generalize to new configurations, such as variations in spatial layouts and object arrangements? To answer these questions, we train and evaluate DINO-WM across five environment suites (full description in Appendix A.1) and compare it to a variety of state-of-the-art world models that model the world both in latent space and in raw pixel space.

### 4.1 Environments and Tasks

We consider five environment suites in our evaluations spanning simple navigation environments and manipulation environments with varying dynamics complexity. For all environments, the observation space is RGB images of size (224, 224). See Appendix A.1 for full details.

a) **Point Maze:** A 2D navigation task from D4RL (Fu et al., 2021), where a point agent with 2D actions navigates a U-shaped maze. The agent's dynamics include velocity, acceleration, and inertia, creating realistic movement. The task requires reaching arbitrary goal locations from arbitrary starting points.

b) **Push-T:** Introduced in (Chi et al., 2024), this environment features a pusher agent interacting with a T-shaped block. The goal is to guide both the agent and the T-block from a randomly initialized state to a known feasible target configuration within 25 steps. The fixed green T serves as a visual reference but not a goal. A variant with multiple object shapes is also explored.

c) **Wall:** This custom 2D navigation environment featuring two rooms separated by a wall with a door opening. The task requires the agent to navigate from a randomized starting location in one room to a goal in the other room, which requires the agent to pass through the door. We introduce

a variant of this environment where the positions of the wall and door are randomized to assess the model's ability to generalize to novel configurations of familiar environment dynamics.

d) **Rope Manipulation:** This task is simulated with Nvidia Flex (Zhang et al., 2024) and consists of an XArm interacting with a rope placed on a tabletop. The objective is to move the rope from an arbitrary start configuration to a specified goal configuration.

e) **Granular Manipulation:** Granular manipulation uses the same setting as Rope manipulation and manipulates about a hundred particles to form desired shapes.

## 4.2 BASELINES

We compare DINO-WM with the following state-of-the-art models commonly used for control:

a) **DreamerV3 (Hafner et al., 2024):** DreamerV3 learns a world model to interpret visual inputs into categorical representation. It predicts future representations and rewards based on given action and trains an actor-critic policy from its imagined trajectories. In our experiments, we train Dreamerv3 agents with our offline datasets without any reward or task information, and perform MPC on the learned world model for solving downstream tasks.

b) **TD-MPC2 (Hansen et al., 2024) :** TD-MPC2 learns a decoder-free world model in latent space and uses reward signals to optimize the latents. It serves as a strong baseline for reconstruction-free world modeling. In our experiments, we train TD-MPC2 agents with our offline datasets without any reward or task information, and perform MPC on the learned world model for solving downstream tasks.

c) **AVDC (Ko et al., 2023):** AVDC leverages a diffusion model to generate an imagined video of task execution based on initial observation and a textual goal description. It then estimates optical flow between frames to capture object movements and generates robot arm commands. Since this model generates an entire sequence of observations conditioned on a given goal, we provide qualitative evaluations of the method, and provide MPC planning results for an action-conditioned variant of the method in Appendix A.7.

## 4.3 OPTIMIZING BEHAVIORS WITH DINO-WM

With a trained world model, we study if DINO-WM can be used for zero-shot planning directly in the latent space. For the PointMaze, Push-T, and Wall environments, we sample 50 initial and goal states to measure the success rate across all instances. Due to the environment stepping time for the Rope and Granular environments, we evaluate the Chamfer Distance (CD) on 10 instances for them. In the Granular environment, we sample a random configuration from the validation set, with the goal of pushing the materials into a square shape at a randomly selected location and scale.

Table 1: Planning results for offline world models on five control environments.

| Model | PointMaze SR ↑ | PushT SR ↑ | Wall SR ↑ | Rope CD ↓ | Granular CD ↓ |
|---|---|---|---|---|---|
| DreamerV3 | **1.00** | 0.04 | **1.00** | 2.49 | 1.048 |
| TD-MPC2 | 0.00 | 0.00 | 0.00 | 2.52 | 1.21 |
| Ours | 0.98 | **0.90** | 0.96 | **0.41** | **0.26** |

As seen in Table 1, on simpler environments such as Wall and PointMaze, DINO-WM is on par with state-of-art world models like DreamerV3. However, DINO-WM significantly outperforms prior work at manipulation environments where rich contact information and object dynamics need to be accurately inferred for task completion. We notice that for TD-MPC2, the lack of reward signal makes it difficult to learn good latent representations, which subsequently results in poor performance. Visualizations of some planning results can be found in Figure 5.

## 4.4 DOES PRE-TRAINED VISUAL REPRESENTATIONS MATTER?

We use different pre-trained general-purpose encoders as the observation model of the world model, and evaluate their downstream planning performance. Specifically, we use the following encoders

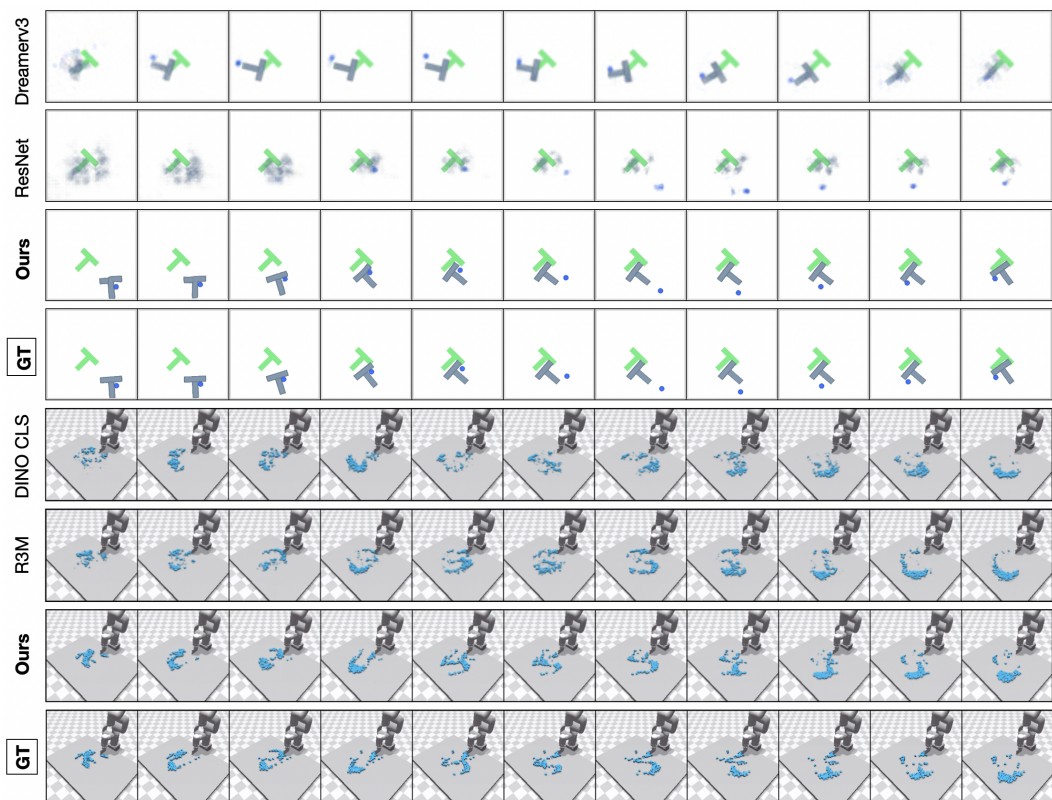

Figure 4: Openloop rollout of world models trained with various pre-trained encoders on Push-T and Granular environment. For each trajectory, the model is given the first frame as well as sequence of actions. The world models performs openloop rollout with these actions, and the images are reconstructed by a pre-trained decoder. For each environment, the bottom row denotes the ground truth. DINO-WM (Ours) rollouts are bolded and are visually indistinguishable from the ground truth observations.

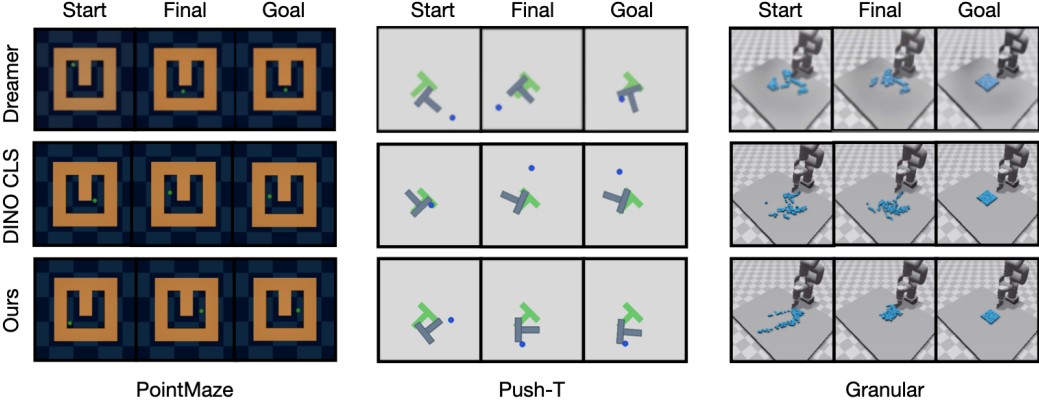

Figure 5: Planning visualizations for PointMaze, Push-T, and Granular, on randomly sampled initial and goal configurations. The task is defined by Start and Goal, denoting the initial and goal observations. Final shows the final state the system arrives at after planning with each world model. For comparison, we show the best performing world models DINO CLS and DreamerV3.

commonly used in robotics control and general perception: R3M (Nair et al., 2022), ImageNet pretrained ResNet-18 (Russakovsky et al., 2015; He et al., 2016) and DINO CLS (Caron et al., 2021). Detailed descriptions of these encoders are in Appendix A.3.

In the PointMaze task, which involves simple dynamics and control, we observe that world models with various observation encoders all achieve near-perfect success rates. However, as the envi-

Table 2: Planning results for world models with various pre-trained encoders.

| Model | PointMaze SR ↑ | PushT SR ↑ | Wall SR ↑ | Rope CD ↓ | Granular CD ↓ |
|---|---|---|---|---|---|
| R3M | 0.94 | 0.42 | 0.34 | 1.13 | 0.95 |
| ResNet | **0.98** | 0.2 | 0.12 | 1.08 | 0.90 |
| DINO CLS | 0.96 | 0.44 | 0.58 | 0.84 | 0.79 |
| DINOPatch (Ours) | **0.98** | **0.90** | **0.96** | **0.41** | **0.26** |

ronment's complexity increases—requiring more precise control and spatial understanding—world models that encode observations as a single latent vector show a significant drop in performance. We posit that patch-based representations better capture spatial information, in contrast to models like R3M, ResNet, and DINO CLS, which reduce observations to a single global feature vector, losing crucial spatial details necessary for manipulation tasks.

## 4.5 GENERALIZING TO NOVEL ENVIRONMENT CONFIGURATIONS

We would like to measure the generalization capability of our world models not just across different goals in an environment, but across different environments configurations. For this we construct three families of environments, where the world model will be deployed in an environment with unseen configurations for randomly sampled goals. Our families of environments consist of Wall-Random, PushObj, and GranularRandom with detailed descriptions in Appendix A.2. Visualizations of training and testing examples are shown in Figure 7.

Table 3: Planning results for offline world models on three suites with unseen environment configurations.

| Model | WallRandom SR ↑ | PushObj SR ↑ | GranularRandom CD ↓ |
|---|---|---|---|
| Dreamerv3 | 0.76 | 0.18 | 1.53 |
| R3M | 0.40 | 0.16 | 1.12 |
| ResNet | 0.40 | 0.14 | 0.98 |
| DINO CLS | 0.64 | 0.18 | 1.36 |
| Ours | **0.82** | **0.34** | **0.63** |

From Table 6, we observe that DINO-WM demonstrates significantly better performance in the WallRandom environment, indicating that the world model has effectively learned the general concepts of walls and doors, even when they are positioned in locations unseen during training. In contrast, other methods struggle to accurately identify the door's position and navigate through it. The PushObj task remains challenging for all methods, as the model was only trained on the four object shapes, which makes it difficult to infer physical parameters like the center of gravity and inertia precisely. In GranularRandom, the agent encounters fewer than half the particles present during training, resulting in out-of-distribution images compared to the training instances. Nevertheless, DINO-WM accurately encodes the scene and successfully gathers the particles into a designated square location with the lowest Chamfer Distance (CD) compared to the baselines, demonstrating better scene understanding. We hypothesize that this is due to DINO-WM's observation model encoding the scene as patch features, making the variance in particle number still within the distribution for each image patch.

## 4.6 QUALITATIVE COMPARISONS WITH GENERATIVE VIDEO MODELS

Given the prominence of generative video models, it is reasonable to presume that they could readily serve as world models. To investigate the usefulness of DINO-WM over such video generative models, we compare it with imagined rollouts from AVDC (Ko et al., 2023), a diffusion-based generative model. As seen in Figure 6, we find that the diffusion model trained on benchmarks produce future images that are mostly visually realistic, however they are not physically plausible as we can see that large changes can occur in a single timestep of prediction, and may have difficulties

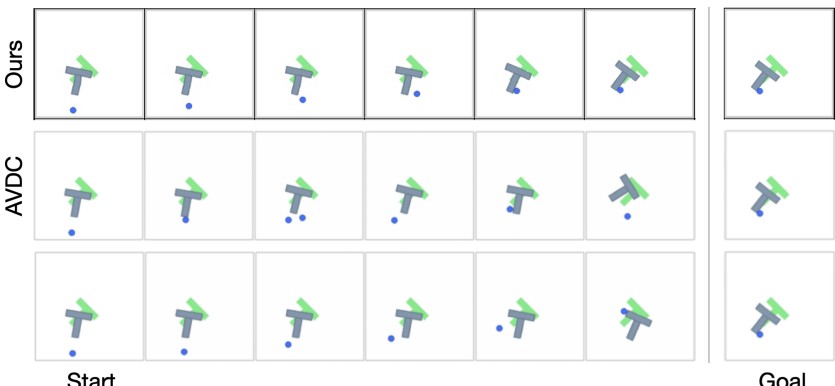

Figure 6: Comparison of plans generated by DINO-WM and AVDC, a diffusion-based generative model.

in reaching to the exact goal state. Perhaps stronger generative models in the future could alleviate this issue.

### 4.7 DECODING AND INTERPRETING THE LATENTS

Although DINO-WM operates in latent space and the observation model is not trained with pixel reconstruction objectives, training a decoder is still valuable for interpreting the model's predictions. We evaluate the image quality of predicted futures across all models and find that our approach outperforms others, even those whose encoders are trained with environment-specific reconstruction objectives. This demonstrates the robustness of DINO-WM despite the lack of explicit pixel-level supervision. We report two key metrics: Structural Similarity Index (SSIM) (Wang et al., 2004) and Learned Perceptual Image Patch Similarity (LPIPS) (Zhang et al., 2018) on the reconstruction of world models' predicted future states. SSIM measures the perceived quality of images by evaluating structural information and luminance consistency between predicted and ground-truth images, with higher values indicating greater similarity. LPIPS, on the other hand, assesses perceptual similarity by comparing deep representations of images, with lower scores reflecting closer visual similarity.

Table 4: Comparison of world models across different environments on LPIPS and SSIM metrics.

| Method | LPIPS ↓ | | | | SSIM ↑ | | | |
|---|---|---|---|---|---|---|---|---|
| | PushT | Wall | Rope | Granular | PushT | Wall | Rope | Granular |
| R3M | 0.045 | 0.0083 | 0.023 | 0.08 | 0.956 | 0.994 | 0.982 | 0.917 |
| ResNet | 0.063 | 0.0024 | 0.025 | 0.08 | 0.950 | 0.996 | 0.980 | 0.915 |
| DinoCLS | 0.039 | 0.004 | 0.029 | 0.086 | 0.973 | 0.996 | 0.980 | 0.912 |
| AVDC | 0.046 | 0.030 | 0.060 | 0.106 | 0.959 | 0.983 | 0.979 | 0.909 |
| Ours | **0.007** | **0.0016** | **0.009** | **0.035** | **0.985** | **0.997** | **0.985** | **0.940** |

## 5 CONCLUSION, LIMITATIONS & FUTURE WORK

In this work, we introduce DINO-WM, a simple yet effective technique for modeling visual dynamics in latent space without the need for pixel-space reconstruction. We have demonstrated that DINO-WM captures environmental dynamics and generalizes to unseen configurations, independent of task specifications, enabling visual reasoning at test time and generating zero-shot solutions for downstream tasks through planning. DINO-WM takes a step toward bridging the gap between task-agnostic world modeling and reasoning and control, offering promising prospects for generic world models in real-world applications. For limitations, DINO-WM still relies on the availability of ground truth actions from agents, which may not always be feasible when training with vast video data from the internet. Additionally, while we currently plan in action space for downstream task solving, an extension of this work could involve developing a hierarchical structure that integrates high-level planning with low-level control policies to enable solving more fine-grained control tasks.

## ETHICS STATEMENT

This work explores creation of latent world models that can be used for better downstream planning. While we do not anticipate a potential for current misuse as this particular work, we can imagine future work that builds on this can lead to impact in robotics. Such potential applications to robotics open up a potential to misuse, which we acknowledge.

## REPRODUCIBILITY STATEMENT

All code, models, and benchmarks produced from this project will be made open-source on our project website. We also provide thorough textual descriptions of all experimental procedures in the Appendix. Appendix A.1 describes our environments, data generation, and task definitions. Appendix A.6 and A.6.3, we outline all the planning optimization methods that we used in this paper. Finally, Appendix A.9 provides the hyperparameters we used for training the world model for reproducing our experiment results in Section 4.1

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

## A    APPENDIX

### A.1    ENVIRONMENTS AND DATASET GENERATION

a) **Point Maze:**    In this environment introduced by Fu et al. (2021), the task is for the 2-DoF ball which is force-actuated in the Cartesian directions x and y to reach a target goal. The agent's dynamics incorporate physical properties such as velocity, acceleration, and inertia, making the movement realistic. We customize the environment by altering the maze configuration to test the model's generalization ability in unseen situations. We generate 2000 fully random trajectories to train our world models.

b) **Push-T:**    In this environment introduced by Chi et al. (2024), the goal is to push a T-shaped block to a designated target position. It has a two-dimensional action space with end-effector position control. Additionally, we introduce variations by altering the shape and color of the object to assess the model's capability to adapt to novel tasks. We generate a dataset of 18500 samples replaying the original released expert trajectories with various levels of noise and evaluate the model's performance across all different shapes to assess its adaptability.

c) **Wall:** This custom 2D navigation environment features two rooms separated by a wall with a door. The agent's task is to navigate from a randomized starting location in one room to a goal in the other, passing through the door. We present a variant where wall and door positions are randomized, testing the model's generalization to novel configurations. For the fixed wall setting, we train on a fully random dataset of 2000 trajectories each with 50 time steps. For the variant with multiple training environment configurations, we generate 10240 random trajectories.

d) **Rope Manipulation:** Described in Zhang et al. (2024), this task is simulated with Nvidia Flex and consists of an XArm interacting with a soft rope placed on a tabletop. The objective is to move the rope from an arbitrary starting configuration to a specified goal configuration. For training, we generate a random dataset of 1000 trajectories of 20 time steps of random actions from random starting positions, while testing involves goal configurations set from varied initial positions, incorporating random variations in orientation, length, and spatial displacement.

e) **Granular Manipulation:** This environment uses the same simulation setup as Rope Manipulation and involves manipulating about a hundred particles to form desired shapes. The training data consists of 1000 trajectories of 20 time steps of random actions starting from the same initial configuration, while testing is performed on specific goal shapes from diverse starting positions, along with random variations in particle distribution, spacing, and orientation.

### A.2    ENVIRONMENT FAMILIES FOR TESTING GENERALIZATION

1. **WallRandom:**    Based on the Wall environment, but with randomized wall and door positions. At test time, the task requires navigating from a random starting position on one side of the wall to a random position on the other side, with non-overlapping wall and door positions seen during training.

2. **PushObj:**    Derived from the Push-T environment, where we introduce novel block shapes, including Tetris-like blocks and a "+" shape. We train the model with four shapes and evaluate on two unseen shapes. The task involves both the agent and object reaching target locations.

3. **GranularRandom:**    Derived from the Granular environment, where we initialize the scene with different amount of particles. The task requires the robot to gather all particles to a square shape at a randomly sampled location. For this task, we directly take the model that is trained with a fixed amount of materials used in Section 4.3.

Visualizations can be found in Figure 7.

Figure 7: Training and testing visualizations for WallRandom, PushObj and GranularRandom. Test setups are highlighted in blue boxes, showcasing unseen configurations for assessing the model's generalization ability.

## A.3 PRETRAINING FEATURES

a) **R3M:** A ResNet-18 model pre-trained on a wide range of real-world human manipulation videos by Nair et al. (2022).

b) **ImageNet:** A ResNet-18 model pre-trained on the ImageNet-1K dataset by Russakovsky et al. (2015).

c) **DINO CLS:** The pre-trained DINOv2 model provides two types of embeddings: Patch and CLS. The CLS embedding is a 1-dimensional vector that encapsulates the global information of an image.

## A.4 EVALUATIONS ON ADDITIONAL BENCHMARKS

**DeepMind Control Suite.** We present additional world model training and planning results on the DeepMind Control Suite (Tassa et al., 2018) to further demonstrate the general applicability of DINO-WM. Specifically, we focus on the Reacher-Hard task, which involves controlling a robotic arm with two joints to reach a target position. To enhance the challenge, we develop a multi-goal version of the task where the agent must reach any goal state from any initial state. This version removes the visual cue (a red dot) indicating the target location, and additionally requires the arm to achieve a precise state, incorporating both the end-effector position and joint angles, instead of merely reaching the correct end-effector position. To train the world models, we collect 3000 randomly generated trajectories each with 100 time steps. Similar as other benchmarks introduced in this task, this dataset does not have any reward information.

In Table 5, we report the performance of DINO-WM along with DreamerV3, our most competitive baseline. We show the success rate of MPC with CEM, as well as a No-Replan success rate, which executes a whole sequence of planned actions once without replan or any feedback from the environment. We show visualizations of the no-replan version in Figure 8. It can be seen that DINO-WM has more accurate predictions over long horizons, and it is able to solve the task without any online interactions or reward information.

Table 5: Planning results for Dreamerv3 and DINO-WM on the Reacher-Hard task in DMControl. Each setting is evaluated on 50 planning instances.

| Model | MPC Success Rate ↑ | No-Replan Success Rate ↑ |
|---|---|---|
| Dreamerv3 | 0.64 | 0.12 |
| Ours | 0.92 | 0.62 |

**LIBERO.** We explore applying DINO-WM to LIBERO, a benchmark targeting life-long robot learning and imitation learning by Liu et al. (2023). Comparing to the deformable environment suite, this benchmark lacks a high-level action space, and the target behavior is highly specific (e.g. open the drawer located at a specific location). This makes it more practical to model such behaviors directly from expert datasets rather than reasoning through a learned world model from

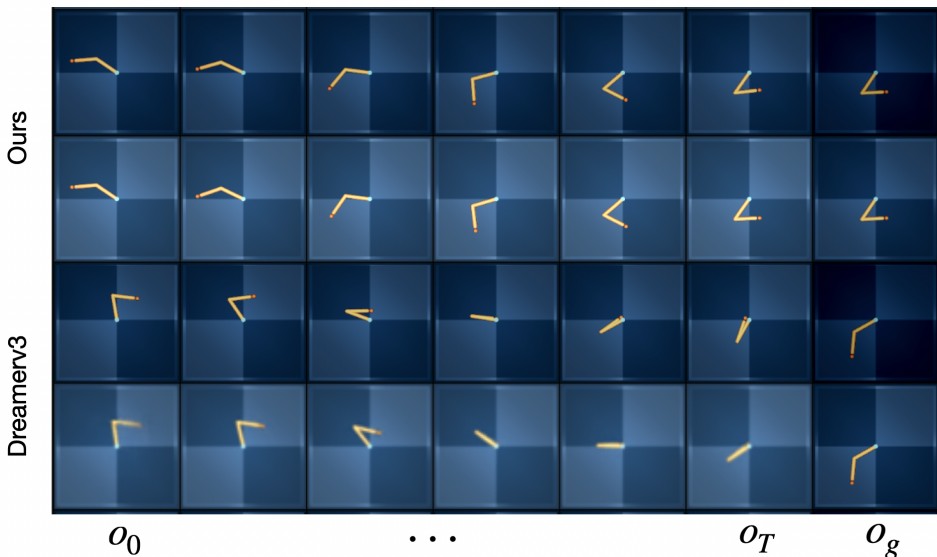

Figure 8: Visualizations of trajectories planned with DINO-WM and Dreamerv3. For each method, the bottom row denotes the world models' predicted rollout reaching the goal observation showing in the right-most column, and the top row (shaded) denotes the environment's actual observation after executing the planned action trajectory. The left-most column denotes the initial observations, and the right-most shaded column denotes the goal observations.

scratch through planning approaches. Naively performing MPC in the world model would lead to exploiting the deficiencies in the world model.

To address this, we introduce another application of the world model: distinguishing expert vs. non-expert trajectories by ranking them based on their predicted quality in solving the task. In this setting, the world model is trained on expert trajectories of 10 tasks of the *libero_goal* suite. At inference time, a collection of expert action trajectories with varying amount of added noise are provided along with an initial observation and a goal observation following the standard planning task setting. We rollout each candidate trajectory through the world model, and obtain planning costs of each trajectory calculated by Equation 3. As shown in Figure 9, it can be seen that the planning costs increase as the trajectories deviate from the expert trajectory, and the learned world model is still able to predict the outcome of such noisy trajectories with reasonable accuracy. This makes DINO-WM a promising approach for integration with multi-task or goal-conditioned policies, where the world model facilitates high-level goal specification predictions. This allows for the chaining of policy executions, effectively guiding an agent to complete tasks based on evolving goals or multi-objective requirements.

### A.5 ABLATIONS

#### A.5.1 SCALING LAWS OF DINO-WM

To analyze the scaling behavior of DINO-WM, we trained world models and performed planning using datasets of varying sizes, ranging from 200 to 18500 trajectories on the PushT environment. Our results demonstrate a clear trend: as the dataset size increases, both the quality of the learned world model and the performance of the planned behavior improve significantly. Larger datasets enable the world model to capture more diverse dynamics and nuances of the environment, leading to more accurate predictions and better-informed planning.

#### A.5.2 DINO-WM WITH VS. WITHOUT CAUSAL ATTENTION MASK

We introduce a causal attention mask in Section 3.1.2. We ablate this choice on PushT by training DINO-WM with and without this causal attention mask with varying history length $h$, such that the model takes in input $o_{t-h+1}, o_{t-h+2}, ...o_t$, and output $o_{t-h+2}, ...o_{t+1}$. For models *with mask*,

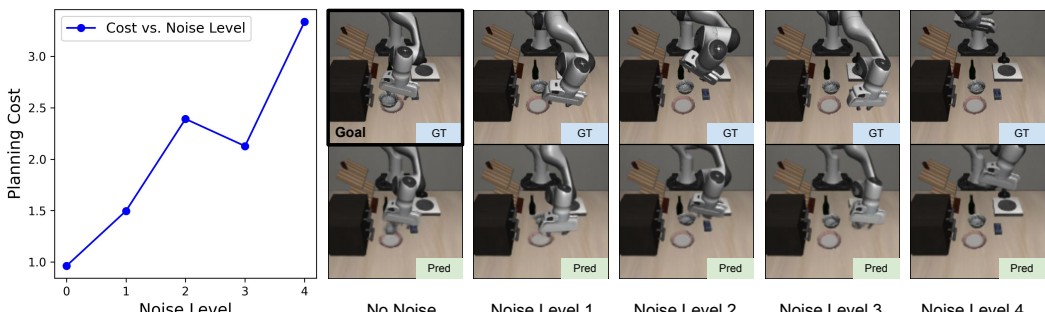

Figure 9: We evaluate expert trajectories with varying levels of noise (scale 1 to 4) added and measure the corresponding planning costs. The left plot shows an increase in planning cost as more noise is added, with the world model accurately predicting the divergence from the original final state. The images on the right visualize the ground truth (GT) final frame and the predicted final frame (Pred) after executing the noisy trajectories. The image labeled with "Goal" represents the target observation for computing the planning cost. Despite being trained with multi-task expert data, the world model successfully predicts outcomes of noisy actions with reasonable accuracy.

Table 6: Planning performance and prediction quality on PushT with DINO-WM trained on dataset of various sizes. SSIM and LPIPS are measured on the predicted future latents after decoding. We observe consistent improvement in performance as we increase the dataset size.

| Dataset Size | SR ↑ | SSIM ↑ | LPIPS ↓ |
|---|---|---|---|
| n=200 | 0.08 | 0.949 | 0.056 |
| n=1000 | 0.48 | 0.973 | 0.013 |
| n=5000 | 0.72 | 0.981 | 0.007 |
| n=10000 | 0.88 | 0.984 | 0.006 |
| n=18500 | 0.92 | 0.987 | 0.005 |

the model can only attend to past observations for predicting each $o_t$, whereas in the *w/o mask* case, predicting any observation in the output sequence can attend to the entire input sequence of observations. We show planning success rate on our PushT settings in Table 7. When $h = 1$ where the model with and without this causal mask is equivalent, both models get decent and equivalent success rate. However, as we increase the history length, we see a rapid drop in the *w/o mask* case, since the model can cheat during training by attending to future frames, which is not available at test time. Adding the causal mask solves this issue, and we observe improvement in performance as longer history could better capture information like velocity, acceleration, and object momentum.

Table 7: Comparison of DINO-WM with and without causal attention mask on PushT. We train models with varying $h$, representing the number of past observations the model takes as input.

| | $h = 1$ | $h = 2$ | $h = 3$ |
|---|---|---|---|
| w/o mask | 0.76 | 0.36 | 0.08 |
| with mask | 0.76 | 0.88 | 0.92 |

### A.5.3 DINO-WM WITH RECONSTRUCTION LOSS

While DINO-WM eliminates the need to train world models with a pixel reconstruction loss—avoiding the risk of learning features irrelevant to downstream tasks—we conduct an ablation study where the predictor is trained using a reconstruction loss propagated from the decoder. As shown in table 8, this approach performs reasonably well on the PushT task but falls slightly short of the proposed version, where the predictor is trained entirely independently of the decoder. This underscores the advantage of disentangling feature learning from reconstruction objectives.

Table 8: Comparison of DINO-WM trained with and without loss from the decoder, highlighting the advantage of disentangling feature learning from reconstruction objectives.

|                  | Success Rate |
| ---------------- | ------------ |
| w/o decoder loss | 0.92         |
| with decoder loss | 0.80        |

## A.6 PLANNING OPTIMIZATION

In this section, we detail the optimization procedures for planning in our experiments.

### A.6.1 MODEL PREDICTIVE CONTROL WITH CROSS-ENTROPY METHOD

a) Given the current observation $o_0$ and the goal observation $o_g$, both represented as RGB images, the observations are first encoded into latent states:

$$\hat{z}_0 = \text{enc}(o_0), \quad z_g = \text{enc}(o_g). \tag{4}$$

b) The planning objective is defined as the mean squared error (MSE) between the predicted latent state at the final timestep $T$ and the goal latent state:

$$\mathcal{C} = \|\hat{z}_T - z_g\|^2, \quad \text{where} \quad \hat{z}_t = p(\hat{z}_{t-1}, a_{t-1}), \quad \hat{z}_0 = \text{enc}(o_0). \tag{5}$$

c) At each planning iteration, CEM samples a population of $N$ action sequences, each of length $T$, from a distribution. The initial distribution is set to be Gaussian.

d) For each sampled action sequence $\{a_0, a_1, \ldots, a_{T-1}\}$, the world model is used to predict the resulting trajectory in the latent space:

$$\hat{z}_t = p(\hat{z}_{t-1}, a_{t-1}), \quad t = 1, \ldots, T. \tag{6}$$

And the cost $\mathcal{C}$ is calculated for each trajectory.

e) The top $K$ action sequences with the lowest cost are selected, and the mean and covariance of the distribution are updated accordingly.

f) A new set of $N$ action sequences is sampled from the updated distribution, and the process repeats until success is achieved or after a fixed number of iterations that we set as hyperparameter.

g) After the optimization process is done, the first $k$ actions $a_0, \ldots a_k$ is executed in the environment. The process then repeats at the next time step with the new observation.

### A.6.2 GRADIENT DESCENT:

Since our world model is differentiable, we also consider an optimization approach using Gradient Descent (GD) which directly minimizes the cost by optimizing the actions through backpropagation.

a) We first encode the current observation $o_0$ and goal observation $o_g$ into latent spaces:

$$\hat{z}_0 = \text{enc}(o_0), \quad z_g = \text{enc}(o_g). \tag{7}$$

b) The objective remains the same as for CEM:

$$\mathcal{C} = \|\hat{z}_T - z_g\|^2, \quad \text{where} \quad \hat{z}_t = p(\hat{z}_{t-1}, a_{t-1}), \quad \hat{z}_0 = \text{enc}(o_0). \tag{8}$$

c) Using the gradients of the cost with respect to the action sequence $\{a_0, a_1, \ldots, a_{T-1}\}$, the actions are updated iteratively:

$$a_t \leftarrow a_t - \eta \frac{\partial \mathcal{C}}{\partial a_t}, \quad t = 0, \ldots, T-1, \tag{9}$$

where $\eta$ is the learning rate

d) The process repeats until a fixed number of iteractions is reached, and we execute the first $k$ actions $a_0, \ldots, a_k$ in the enviornment, where $k$ is a pre-determined hyperparameter.

### A.6.3 PLANNING RESULTS

Here we present the full planning performance using various planning optimization methods. CEM denotes the setting where we use CEM to optimize a sequence of actions, and execute those actions in the environment without any correction or replan. Similarly, GD denotes optimizing with gradient decent and execute all planned actions at once in an open-loop way. MPC denotes allowing replan and receding horizon with CEM for optimization.

Table 9: Planning results of DINO-WM

|     | PointMaze | Push-T | Wall | Rope | Granular |
| --- | --- | --- | --- | --- | --- |
| CEM | 0.8 | 0.86 | 0.74 | NA | NA |
| GD  | 0.22 | 0.28 | NA | NA | NA |
| MPC | 0.98 | 0.90 | 0.96 | 0.41 | 0.26 |

### A.7 COMPARISON WITH ACTION-CONDITIONED GENERATIVE MODELS

We compare DINO-WM with a variant of AVDC, where the diffusion model is trained to generate the next observation $o_{t+1}$ conditioned on the current observation $o_t$ and action $a_t$, rather than generating an entire sequence of observations at once conditioned on a text goal. We then present open-loop rollout and planning results on validation trajectories using this action-conditioned diffusion model, with visualizations shown in Figure 10. It can be seen that the action-conditioned diffusion model diverges from the ground truth observations over long-term predictions, making it insufficient for accurate task planning. This is further corroborated by our planning experiments, where the action-conditioned AVDC model achieves a planning success rate of 0% on PushT, demonstrating its inadequacy for the intended tasks.

### A.8 INFERENCE TIME

Inference time is a critical factor when deploying a model for real-time decision-making. Table 10 reports the time required for a single inference step, the environment rollout time for advancing one step in the simulator, and the overall planning time for generating an optimal action sequence using the Cross-Entropy Method (CEM). The inference time of DINO-WM remains constant across environments due to the fixed model size and input image resolution, resulting in significant speedup over traditional simulation rollouts. Notably, in environments with high computational demands, such as deformable object manipulation, simulation rollouts require several seconds per step while DINO-WM enables rapid inference and efficient planning. Planning time is measured with CEM using 100 samples per iteration and 10 optimization steps, demonstrating that DINO-WM can achieve feasible planning times while maintaining accuracy and adaptability across tasks.

Table 10: Inference time and planning time for DINO-WM. Inference time represents the time required for a single forward pass for one step, while environment rollout time measures the simulator's speed for advancing one step. Planning time corresponds to Cross-Entropy Method (CEM) with 100 samples per iteration and 10 optimization steps.

| Metric | Time (s) |
| --- | --- |
| Inference (Batch 32) | 0.014 |
| Simulation Rollout (Batch 1) | 3.0 |
| Planning (CEM, 100x10) | 53.0 |

### A.9 HYPERPARAMETERS AND IMPLEMENTATION

We present the DINO-WM hyperparameters and relevant implementation repos below. We train the world models for all environments with the same hyperparameters.

The world model architecture is consistent across all environments. We use an encoder based on DINOv2, which extracts features with a shape of $(14 \times 14, 384)$ from input images resized to $224 \times$

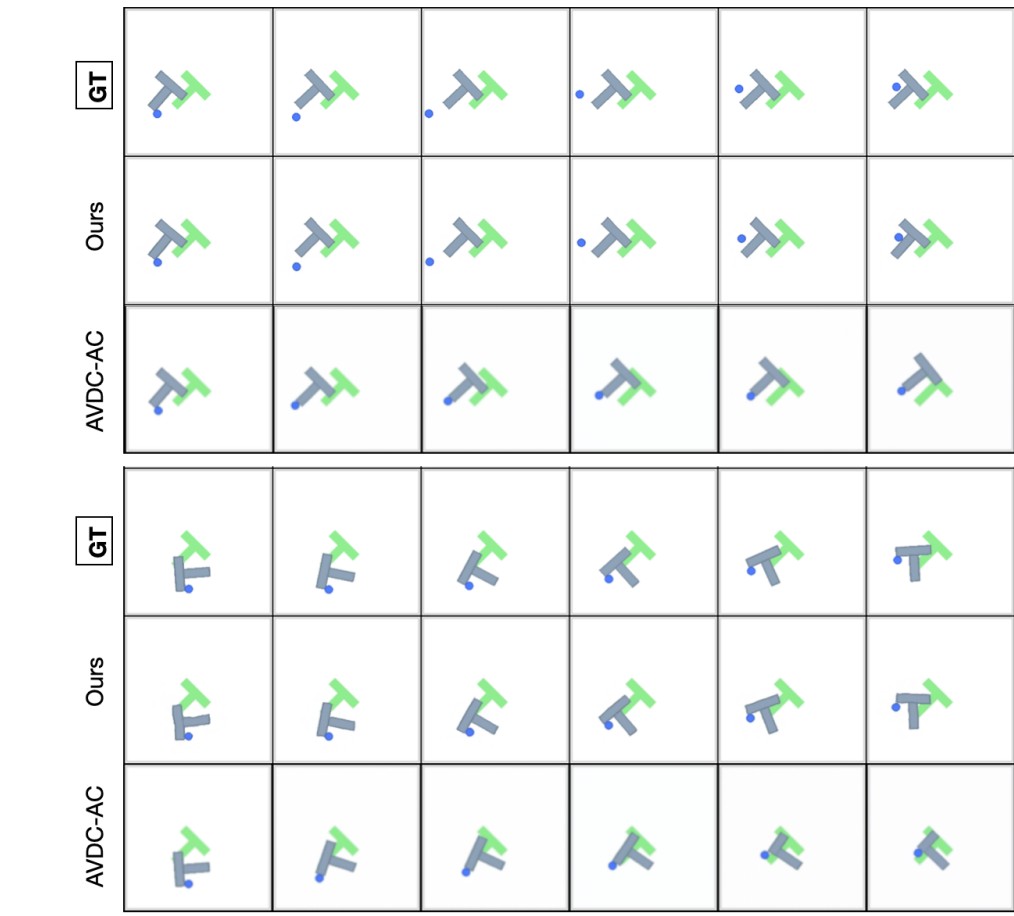

Figure 10: Openloop rollout on PushT with DINO-WM and action-conditioned AVDC (AVDC-AC). For each trajectory, the model is given the first frame as well as sequence of actions. The world models performs openloop rollout with these actions.

224 pixels. The ViT backbone has a depth of 6, 16 attention heads, and an MLP dimension of 2048, amounting to approximately 19M parameters.

Table 11: Environment-dependent hyperparameters for DINO-WM training. We report the number of trajectories in the dataset under *Dataset Size*, and the length of trajectories under *Traj. Len.*

|  | $H$ | Dataset Size | Traj. Len. |
| --- | --- | --- | --- |
| PointMaze | 3 | 2000 | 100 |
| Push-T | 3 | 18500 | 100-300 |
| PushObj | 3 | 20000 | 100 |
| Wall | 1 | 2000 | 100 |
| WallRandom | 1 | 10240 | 100 |
| Rope | 1 | 1000 | 5 |
| Granular | 1 | 1000 | 5 |

Table 12: Shared hyperparameters for DINO-WM training

| Name | Value |
| --- | --- |
| Image size | 224 |
| Optimizer | AdamW |
| Decoder lr | 3e-4 |
| Predictor lr | 5e-5 |
| Action encoder lr | 5e-4 |
| Action emb dim | 10 |
| Epochs | 100 |
| Batch size | 32 |

- DINOv2: https://github.com/facebookresearch/dinov2
- DreamerV3: https://github.com/NM512/dreamerv3-torch
- AVDC: https://github.com/flow-diffusion/AVDC

- R3M: `https://github.com/facebookresearch/r3m/`

We base our predictor implementation on `https://github.com/lucidrains/vit-pytorch/`.

## A.10 ADDITIONAL PLANNING VISUALIZATIONS

We present additional visualizations for planning with DINO-WM. In this setting, all planning instances share the same initial observations but have different goal observations to demonstrate DINO-WM's generalization capabilities in planning. We show trajectory pairs to compare the environment's observations after executing a sequence of planned actions with DINO-WM's imagined trajectories. The left-most column denotes the initial observations, and the right-most shaded column denotes the goal observations. Each pair of rows represents a planning instance: the top (shaded) row shows the environment's observation after executing 25 planned actions, and the bottom row shows the world model's imagined observations.

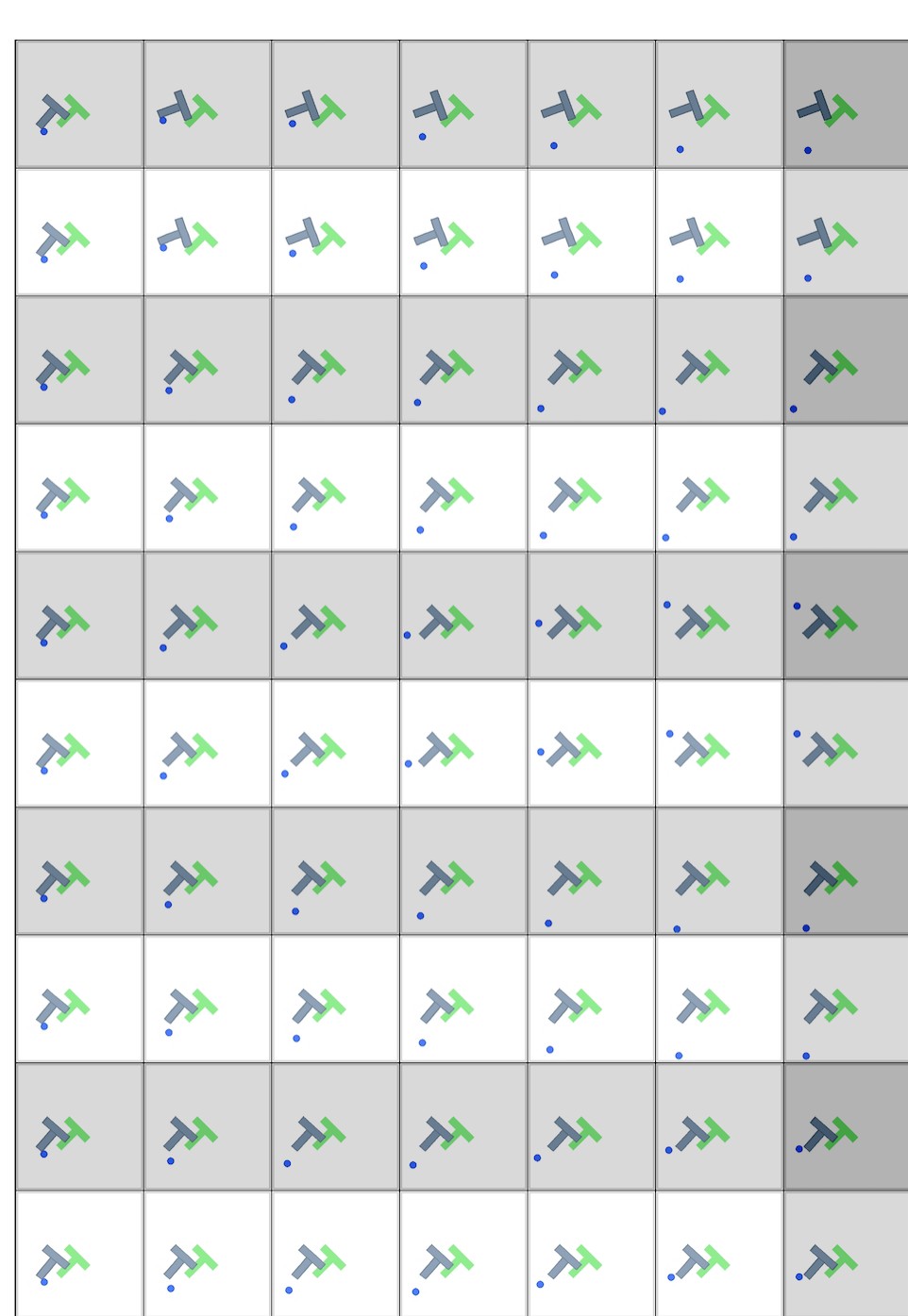

Figure 11: Trajectories planned with DINO-WM on PushT with the same initial states but different goal states.

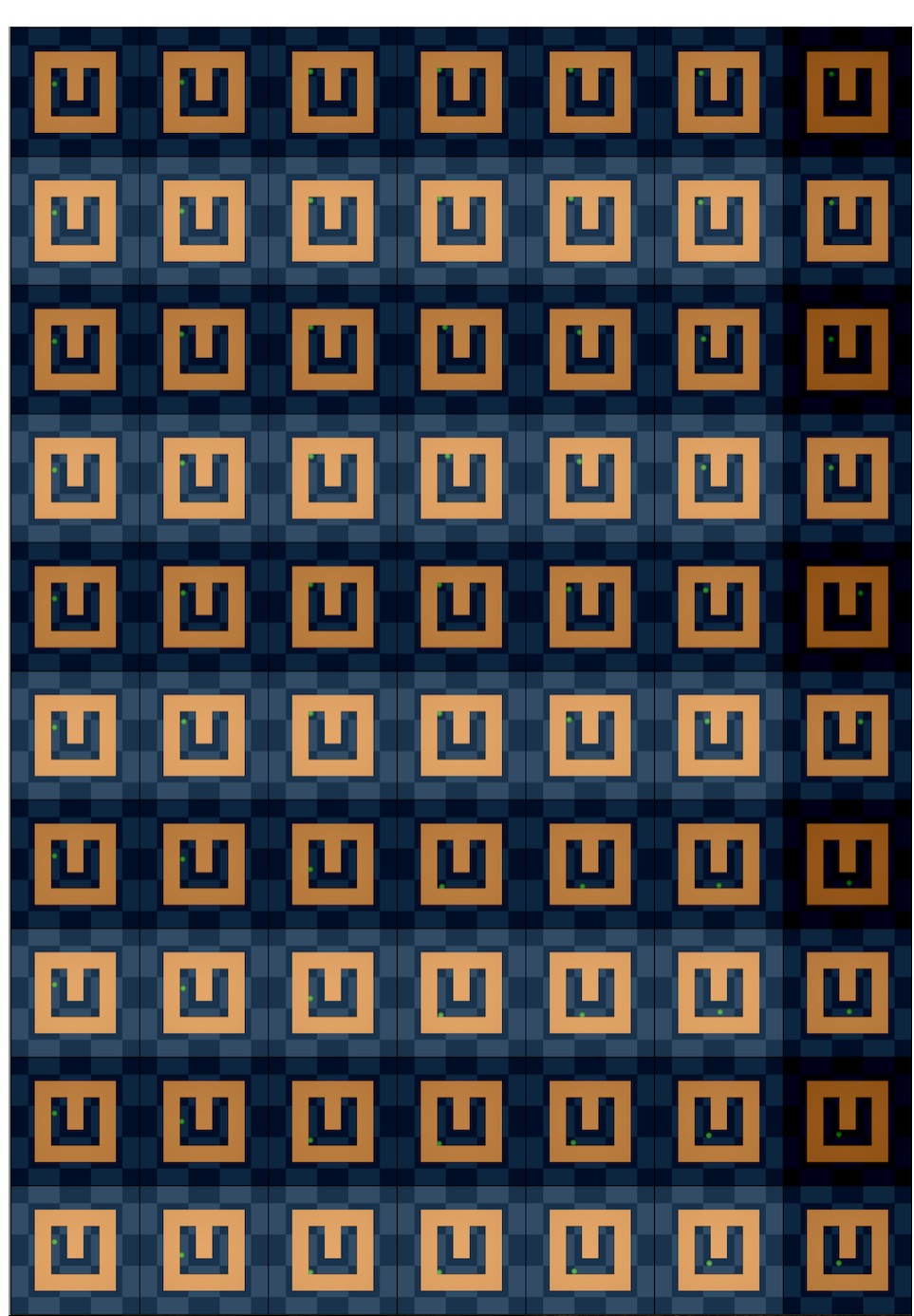

Figure 12: Trajectories planned with DINO-WM on PointMaze with the same initial states but different goal states.

