# OpenReview forum: "DINO-WM: World Models on Pre-trained Visual Features enable Zero-shot Planning"
_ICLR.cc/2025/Conference — Submitted to ICLR 2025_

### Official Review · Reviewer_fU9L · 2024-11-01

**Soundness:** 3
**Presentation:** 3
**Contribution:** 2
**Rating:** 6
**Confidence:** 5

**Summary:**

This work proposes a simple approach for learning a task-agnostic world model from offline data. They use frozen DINOv2 tokens and a Transformer transition model with a single-step forward prediction loss. This world model is then shown to enable model predictive control in toy environments as well as two simulated robotic environments involving manipulating a rope and many particles.

**Strengths:**

1. The authors demonstrate that frozen visual tokens from DINOv2 are enough to learn a forward model that successfully predicts the dynamics of several different robot control environments.
2. It is further shown that the world model is able to plan in altered environments because, in part, it does not depend on a task-specific reward function.
3. The proposed world model compares favorably to task-specific world models, DreamerV3 and TD-MPC2, and its decoder generates more plausible trajectories than a prior video prediction method, AVDC.
4. The rope and granular manipulation environments demonstrate the simulation of difficult transition dynamics.

**Weaknesses:**

1. The paper includes an insufficient set of benchmarks. I would recommend including more standard reinforcement learning and robotics benchmarks, such as DMControl, RoboMimic, MimicGen, RLBench, MetaWorld, FrankaKitchen, ManiSkill, etc. These benchmarks would better assess the world model’s ability to capture robot-object dynamics and sequential tasks. For example, the TD-MPC2 paper (used as a baseline) includes MetaWorld, DMControl, ManiSkill and MyoSuite as benchmarks. It would only be fair to compare the proposed method on some subset of the benchmarks used in TD-MPC2 or DreamerV3. I would recommend adding at least one benchmark from TD-MPC2 to strengthen the comparison. E.g., MetaWorld covers a diverse set of manipulation tasks.
2. The paper focuses on learning task-agnostic world models from offline data, but this entails a major assumption. It is assumed that the training dataset sufficiently covers the environment’s state-action space. This appears to be the case in Point Maze, Two Room, Rope Manipulation and Granular Manipulation, where the datasets are collected by executing random actions. This would not work in, e.g., RoboMimic or FrankaKitchen, where the agent has to execute a sequential task that is difficult to solve with random actions. Here, we have a chicken-and-egg problem. We need an expert dataset to train an accurate world model, but we need an accurate world model to plan expert behavior. This is why prior world modeling approaches focus on online improvement and joint learning of world models and policies.
3. I am confused by the first sentence of Section 3.1.2. It states that the ViT model is used as the transition model. But ViT is just a Transformer with tokenized images and the input to the transition model is already tokenized. Wouldn’t it be more accurate to say that a decoder-only Transformer is used as the transition model?
4. Section 3.1.2 further describes a causal masking approach of treating patch vectors of one observation as a whole unit. This design decision should be ablated in the experiments. In particular, comparing the planning success rate of DINO-WM with and without the per-image causal mask.
5. The comparison with baselines would be more fair if DreamerV3 and TD-MPC2 were modified to also use a DINOv2 image encoder, which I assume is not the case in the current experiments. Could you swap the convolutional encoder of DreamerV3 with DINOv2 and measure the impact on its planning success rate?
6. Embed to Control (https://arxiv.org/abs/1506.07365) is a major missing citation, which also investigates offline world model learning without rewards.

**Questions:**

1. The decoder could propagate gradients into the transition model if you first predicted the next state, then decoded it and compared it to the original next state image? Have you considered this design decision?
2. Some world modeling approaches roll out trajectories N steps into the future to improve the consistency of predictions over time. Would that be compatible with your approach?
3. Do you fine-tune AVDC on your datasets?

---

> ### Author Response · Authors · 2024-11-21
>
> We thank the reviewer for their thorough and insightful reviews, especially in highlighting that the world model is able to “plan in altered environments,” and can handle “simulation of difficult transition dynamics.”
>
> We now address the issues raised by the reviewer.
> # W1: Evaluations on Additional Benchmarks
> > “ I would recommend including more standard reinforcement learning and robotics benchmarks, such as DMControl, RoboMimic, MimicGen, RLBench, MetaWorld, FrankaKitchen, ManiSkill, etc.”“I would recommend adding at least one benchmark from TD-MPC2 to strengthen the comparison.”
>
> We thank the reviewer for this suggestion. Please refer to our global comments on why the standard RL benchmarks may not be suitable for evaluating DINO-WM, as it is not an task-specific online RL algorithm. Additionally, we would like to clarify that our evaluations on baselines like Dreamerv3 and TDMPC2 focuses on evaluating the world model component of each method, which is different from the evaluation proposed in their original papers where they have online interactions with the environment, have access to rewards, and focuses on single task settings.
>
> However, we agree that including evaluations on these benchmarks would further strengthen the comparison and demonstrate DINO-WM’s applicability, thus  we conducted an additional experiment on DMControl’s *Reacher-Hard* task, and further enhanced its difficulty by extending it to a multi-goal version. We compared planning performance between DINO-WM and DreamerV3, our strongest baseline. We observe a 43% performance improvement for DINO-WM compared to DreamerV3. Complete results and additional visualizations are provided in Appendix A.4. Additionally, we include planned trajectory visualizations on our website linked in the manuscript for further reference.
>
> Additionally, we performed further evaluations of DINO-WM on LIBERO[1], a benchmark designed for lifelong robot learning and imitation learning. Here, we demonstrated that DINO-WM, trained on multi-task expert trajectories, effectively distinguishes between expert and non-expert trajectories. This capability positions DINO-WM as a promising approach for integration with multi-task or goal-conditioned policies, enabling the world model to facilitate predictions with high-level goal specifications.

---

> > ### Author Response · Authors · 2024-11-21
> >
> > # W2: Dataset Coverage Assumption
> > > “It is assumed that the training dataset sufficiently covers the environment’s state-action space … This would not work in, e.g., RoboMimic or FrankaKitchen, where the agent has to execute a sequential task that is difficult to solve with random actions. Here, we have a chicken-and-egg problem. We need an expert dataset to train an accurate world model, but we need an accurate world model to plan expert behavior. This is why prior world modeling approaches focus on online improvement and joint learning of world models and policies.”
> >
> > We appreciate the reviewer's insightful observations and agree that sufficient coverage of the environment’s state-action space is crucial for task-agnostic planning to succeed. This requirement is met in our proposed benchmarks, where datasets collected through random actions provide the necessary diversity.
> >
> > For task suites like RoboMimic or FrankaKitchen where random rollouts wouldn’t provide this coverage, we agree that task-agnostic planning with the world model won’t work well, presenting a challenging open question for task-agnostic world modeling. We believe that the solution is not to couple world model training with task-specific objectives, as explored in prior work. While such approaches are effective for solving predefined tasks with reward signals, they are inherently limited to the specific tasks they are trained on. This narrow focus makes it unclear whether these models have any potential for generalizing to other tasks in similar environments. Although they are effective methods within the scope of online RL, their utility as broadly applicable world models remains uncertain.
> >
> > We believe a promising direction to address this challenge is to structure world model predictions hierarchically. Analogous to human planning, which often relies on high-level planning while delegating low-level execution to conditioned policies, future iterations of task-agnostic models like DINO-WM could incorporate hierarchical planning, where high-level reasoning with the world model is paired with lower-level execution modules, making the predictions and search to be more efficient and informative. Hierarchical rollouts could support downstream tasks better than random low-level action rollouts.
> >
> > In this regard, we have included an additional evaluation on the LIBERO[1] benchmark in Appendix A.4 of the updated manuscript, showing that a trained generic world model can be used to identify trajectories that solve a particular task. This makes DINO-WM a promising approach for integration with multi-task or goal-conditioned policies as natural hierarchies, where the world model facilitates high-level goal specification predictions. This allows for the chaining of policy executions, effectively guiding an agent to complete tasks based on evolving goals or multi-objective requirements.
> >
> > As for the online improvement aspects, we fully agree that continuously updating the world model as more data becomes available could significantly enhance its performance, and could be a promising future direction for DINO-WM.
> >
> > # W3: Update Terminologies for the Predictor Architecture
> > > “Wouldn’t it be more accurate to say that a decoder-only Transformer is used as the transition model?”
> >
> > We agree that the architecture of the predictor is essentially a decoder-only transformer since we removed the image tokenization layers typically used in ViTs. However, we used the terminology "ViT" in the manuscript to avoid assumptions typically associated with "decoder-only transformers," such as the use of a causal attention mask at the token level, which differs from the frame-level attention mask we employ. Our intention was to describe the implementation more clearly: we start with the ViT architecture, remove the tokenization layer, and introduce the frame-level causal mask. We have now revised the manuscript for clarity on this distinction.
> >
> >
> > # W4: Ablations on the Causal Attention Mask
> > > “This design decision should be ablated in the experiments. In particular, comparing the planning success rate of DINO-WM with and without the per-image causal mask.”
> >
> > We thank the reviewer for suggesting this ablation experiment. We have now conducted experiments comparing the planning performance of DINO-WM on downstream tasks with and without the per-image causal attention mask, for world models with history lengths ranging from 1 to 3. Our results show that as the history length increases, there is a significant drop in performance when the causal mask is not applied. This occurs because, without the mask, the model can "cheat" during training by attending to future frames, which is not available at planning time. Adding the causal mask mitigates this issue, resulting in improved performance, particularly when longer histories are used to capture dynamics such as velocity, acceleration, and object momentum. Full results have been added to the manuscript in Appendix A.5.2.

---

> > > ### Author Response · Authors · 2024-11-21
> > >
> > > # W5: Using DINO Feature for Baselines
> > > > “Could you swap the convolutional encoder of DreamerV3 with DINOv2 and measure the impact on its planning success rate?”
> > >
> > > Thank you for the suggestion. While we recognize the value of comparing DreamerV3 and TD-MPC2 using a DINOv2 image encoder for a fairer baseline, such modifications are not feasible for the following reasons:
> > > 1. **Task Specifications and the Focus of Our Comparison**:  Our experiments focus specifically on the world model component of DINO-WM and the baseline methods, isolating the dynamics learning process without incorporating task-specific feedback or reward signals. This is important because our primary aim is to compare the learned representations and dynamics and whether they could be effectively used for reasoning for downstream tasks, not the policy optimization or task-solving capabilities that rely on rewards and environment interaction.
> > >
> > >     Therefore, comparing our method directly with the full implementations of TD-MPC2 or DreamerV3, which include task-specific learning components, would not provide a meaningful evaluation. Instead, we isolate and compare only their world model architectures (e.g., RSSMs, convolutional encoders and MLPs) against our DINO-based world model under consistent settings.
> > >
> > > 2. **Architectural Incompatibility**: DreamerV3 and TD-MPC2 are designed to work with CNN or RSSM-based encoders, not the high-dimensional patch representations from DINOv2. Modifying these architectures to incorporate DINOv2 would require substantial changes that would effectively replicate DINO-WM rather than provide a meaningful comparison. For instance, DreamerV3’s use of 32-frame temporal context is computationally prohibitive with DINO’s patch features, whereas we use at most 3-frame temporal context for DINO-WM in our experiments. For TD-MPC2, the latent dynamics model’s architecture is a simple MLP, which is infeasible to be directly applied to work with DINO’s patch features.
> > > Therefore, modifying these baselines would make them essentially versions of DINO-WM, which is not the intention of the comparison.
> > > # W6: Missing Citation
> > > > “Embed to Control (https://arxiv.org/abs/1506.07365) is a major missing citation, which also investigates offline world model learning without rewards.”
> > >
> > > We thank the reviewer for the reference, and we have updated our manuscript to include this citation.
> > >
> > >
> > > We now address the reviewer’s questions.
> > > # Q1: Backprop Decoder Loss to Predictor
> > > > “The decoder could propagate gradients into the transition model if you first predicted the next state, then decoded it and compared it to the original next state image? Have you considered this design decision?”
> > >
> > > While DINO-WM intentionally eliminates the need for a pixel reconstruction loss to avoid the risk of learning features irrelevant to downstream tasks, we conducted an additional experiment to address the reviewer's suggestion. This experiment essentially involves training the predictor with the decoder's reconstruction loss. The results of this experiment are included in Appendix A.5.3 of the manuscript. We observe that this approach performs slightly worse on the PushT task compared to the proposed method, where the predictor is trained independently of the decoder. This highlights the advantage of disentangling feature learning from reconstruction objectives.
> > >
> > > # Q2: Rolling Out Trajectories N Steps into the Future
> > > > “Some world modeling approaches roll out trajectories N steps into the future to improve the consistency of predictions over time. Would that be compatible with your approach?”
> > >
> > > Great point! In fact, we are already doing this in all of our planning experiments with MPC. As described in Section 3.2 and Appendix A.6.1, the planning optimization method we use, CEM, optimizes for a sequence of N actions simultaneously. This process involves rolling out trajectories N steps into the future  in an autoregressive manner with the world model. This also makes it possible to plan successful trajectories without replan or any environment feedback at test time.
> > > # Q3: Finetuning AVDC
> > > > “Do you fine-tune AVDC on your datasets?”
> > >
> > > Yes, we fine-tune AVDC on our datasets for all the AVDC experiments presented in the manuscript. We are happy to answer any additional questions the reviewer may have regarding this.
> > > # Conclusion
> > > We appreciate the reviewer's suggestion for additional experiments. We hope the provided clarifications resolve any outstanding concerns, and we are happy to discuss any further points that stand between us and a higher score.
> > >
> > > # References
> > >
> > > [1] Bo Liu, Yifeng Zhu, Chongkai Gao, Yihao Feng, Qiang Liu, Yuke Zhu, Peter Stone. LIBERO: Benchmarking Knowledge Transfer for Lifelong Robot Learning

---

> > > > ### Comment · Reviewer_fU9L · 2024-11-25
> > > > **Response**
> > > >
> > > > Thank you for addressing my questions and including additional benchmarks and ablations in the appendix! I will raise my score.
> > > >
> > > > I am leaning towards accepting this paper, but I still have reservations about the offline learning setting and the limited novelty of this paper. I believe that general world models will be achieved by combining a large number of task-specific datasets. In contrast, the focus on environments where random behavior leads to meaningful states is inherently limiting.

---

> > > > > ### Author Response · Authors · 2024-11-26
> > > > >
> > > > > Thank you for your thoughtful feedback and for raising your score — we truly appreciate your support and detailed evaluation of our work.
> > > > >
> > > > > Regarding the offline learning setting, we strongly believe this approach is crucial for building general world models. If the assumption is that a large number of task-specific datasets will be used, a robust world model should ideally be able to train effectively using only this offline data without requiring online interaction. This aligns with the vision of leveraging diverse datasets to develop general-purpose world models.
> > > > >
> > > > > On your point about environments where random behavior leads to meaningful states, we acknowledge this limitation but would like to emphasize that the use of higher-level actions, skills, or behavioral priors could address this concern. As shown in prior work [1], even rolling out these skills randomly can lead to meaningful behaviors. We view this as an exciting direction for extending the capabilities of general world models.
> > > > >
> > > > > If you feel our responses and clarifications sufficiently address your reservations, would you consider increasing your score? Regardless, we deeply value your constructive insights and your support for our work.
> > > > >
> > > > > Thank you again for your time and consideration!
> > > > >
> > > > >
> > > > > [1] Avi Singh, Huihan Liu, Gaoyue Zhou, Albert Yu, Nicholas Rhinehart, Sergey Levine. Parrot: Data-Driven Behavioral Priors for Reinforcement Learning

---

### Official Review · Reviewer_Cbfu · 2024-11-03

**Soundness:** 3
**Presentation:** 3
**Contribution:** 3
**Rating:** 6
**Confidence:** 4

**Summary:**

The paper proposes Dino-WM a method to model visual dynamics without reconstructing the visual world by leveraging patch features from pretrained visual encoder DinoV2, allowing it to learn the world model from offline datasets of trajectories. This formulation allows Dino-WM to realize observational goals by using action sequence optimizing enabling task-agnostic behavior planning. The paper demonstrates Dino-WM can generate solutions for tasks at test-time zero-shot without any expert demonstration, reward modeling or inverse dynamics models. In addition, Dino-WM outperforms existing methods like DreamerV3 and TD-MPC2 on complex control environments. Additionally, paper present additional analysis on using various pretrained visual encoder for world modelling to demonstrate effectiveness of using DinoV2 features and show generalization to novel environment configurations.

**Strengths:**

1. The paper is well written and easy to follow
2. The idea proposed in the paper is quite simple, scalable and effective as demonstrated by the results.
3. The experiments section is thorough and clearly demonstrates effectiveness of Dino-WM compared to existing methods.

**Weaknesses:**

1. The authors argue in the intro and abstract of the paper that a world model should enable learning from **only passive data** but the experiments demonstrated in the paper always require action metadata during training of Dino-WM which contradicts the motivation. I’d suggest authors to make that argument a bit less constrained to convey the world model training should include both passive and action conditioned data.
2. The comparisons presented in experiments section 4.1 do not include comparisons to diffusion based methods. I understand it might not be feasible to get results for all environments but I’d recommend authors to add comparison of AVDC or another diffusion based methods for whichever task possible in table.1. If not possible, I’d recommend authors to add a discussion about why the comparison is not possible. Currently, the section 4.2 implies there’ll be quantitative comparison with diffusion based methods.
3. The generalization experiments in section 4.5 tests generalization to unseen configurations of environments seen during training but the description in section 4.5 (L 450-451) implies that experiments test generalization to new environments on which the model was never trained which is not true. I’d recommend authors to please fix the text description in this section.
4. I couldn’t find any training details for world models used in experiments section. It is not clear to me what datasets were used for training Dino-WM, how were the trajectories collected, what was the scale of those datasets,etc.  The appendix has hyperparam details but no training details. It’d be great if authors can add description for those details in the paper.
5. I would also be interested in seeing scaling laws analysis for the a subset of environments presented in the paper. It’d be interesting to see how evaluation performance improves as the size of training dataset used for Dino-WM pretraining was used. I believe it will make paper stronger and would be a nice analysis to have.

**Questions:**

In addition to AVDC there are more recent WM style methods based on generative methods like Genie [1] that work quite well. I'd like authors to either present comparison with this method or add a discussion addressing why the comparison is not possible.

[1] Genie: Generative Interactive Environments

---

> ### Author Response · Authors · 2024-11-21
>
> We thank the reviewer for their thorough and constructive feedback, especially for highlighting that our work DINO-WM is “quite simple, scalable and effective,” and that it “generates solutions for tasks at test-time zero-shot without any expert demonstration, reward modeling or inverse dynamics models.”
>
> We now address the issues raised in the review.
> # Clarification on “Passive Data”
>
> > “The authors argue in the intro and abstract of the paper that a world model should enable learning from only passive data but the experiments demonstrated in the paper always require action metadata during training of Dino-WM which contradicts the motivation.”
>
> We appreciate the reviewer's feedback. To clarify, by "passive data," we refer to pre-collected offline datasets that do not require additional interactions with the environment during training, instead of datasets with no action labels. This distinction from online interaction-based data collection has been explicitly clarified in the revised manuscript’s introduction and abstract to better align with the paper's scope and methodology.
> # Planning Results for Diffusion-Based Baselines
>
> > “ I’d recommend authors to add comparison of AVDC or another diffusion based methods for whichever task possible in table.1”
>
> We note the original AVDC is not compatible with planning evaluations, as it is a goal-conditioned generation model. This model generates entire sequences of observations conditioned on a target goal state, making it incompatible with our tasks, which involve iteratively planning and evaluating actions. To adapt AVDC for planning, one would require an additional inverse dynamics model to convert generated observation sequences into actionable plans. Furthermore, as we visualized in Section 4.6 of the original manuscript, the plans generated by AVDC often fail to reach the specified goal positions, exhibit large discontinuities between states, and sometimes include unrealistic environment configurations. These limitations would result in a task-solving success rate of 0, even if a trained inverse dynamics model were used.
>
> To address the reviewer’s feedback, we have further implemented an action-conditioned variant of AVDC. Instead of generating the entire sequence conditioned on a goal, this variant predicts the next frame given the current frame and action, enabling it to be evaluated in the same planning framework as DINO-WM. However, this approach exhibited severe compounding errors over long horizons, ultimately resulting in a planning success rate of 0% on the PushT task. The full results, including open-loop rollouts and planning evaluations, have been added to Appendix A.7 in the updated manuscript.
>
> > “Currently, the section 4.2 implies there’ll be quantitative comparison with diffusion based methods.”
>
> Thank you for raising this point. We have updated the manuscript to clarify this.
>
> # Clarification for “New Environments”
> > “experiments in section 4.5 tests generalization to unseen configurations of environments seen during training but the description in section 4.5 (L 450-451) implies that experiments test generalization to new environments on which the model was never trained which is not true. I’d recommend authors to please fix the text description in this section.”
>
> We apologize for the confusion in the problem setting in Section 4.5. The experiments indeed focus on testing generalization to unseen configurations of environments encountered during training. For example, in the Wall environment, this involves walls and doors at previously unseen locations; in PushObj, objects of novel shapes and colors; and in the Granular environment, an unseen quantity of granular materials. These experiments do not evaluate generalization to entirely novel environments on which the model was never trained.
>
> We have revised the text in Section 4.5 of the manuscript to clarify the scope of the experiments and ensure the description accurately conveys the intended problem setting. Thank you for bringing this to our attention.
>
> # Training Details for DINO-WM
> > “It is not clear to me what datasets were used for training Dino-WM, how were the trajectories collected, what was the scale of those datasets,etc.”
>
> The dataset size and details about data collection are provided in Appendix A.1 of the original manuscript, including the scale of the datasets and the procedures used to collect trajectories. To address the concern more explicitly, we have updated the manuscript to also include this information in Appendix A.9, where we consolidated training-related details for better accessibility. We have further added the model’s architecture details in Appendix A.9 as well.
>
> If additional specifics regarding training processes, model hyperparameters, or further dataset details would be helpful, we are more than happy to provide them and update the manuscript.

---

> > ### Author Response · Authors · 2024-11-21
> >
> > # Scaling Laws Analysis
> > > “It’d be interesting to see how evaluation performance improves as the size of training dataset used for Dino-WM pretraining was used. I believe it will make paper stronger and would be a nice analysis to have.”
> >
> > We thank the reviewer for this insightful suggestion! In response, we have conducted experiments evaluating DINO-WM's performance on the PushT task using varying training dataset sizes, from 200 samples up to the full dataset of 18,500 samples. The results, detailed in Appendix A.5.1 in the updated manuscript, include metrics such as planning success rate, SSIM, and LPIPS for predicted future frames. These experiments reveal a clear and consistent trend: as the dataset size increases, both the quality of the learned world model and the performance of the planning significantly improve. This analysis highlights the scalability of DINO-WM and its ability to leverage larger datasets for better generalization and task-solving capabilities.
> >
> > We now answer the reviewer’s questions.
> > # Comparing with Genie
> >
> > > In addition to AVDC there are more recent WM style methods based on generative methods like Genie [1] that work quite well. I'd like authors to either present comparison with this method or add a discussion addressing why the comparison is not possible.
> >
> >
> > Thank you for the suggestion. Genie [1] differs fundamentally from our work in the purpose and design. It is a foundation model for generating open-ended, interactive environments, prioritizing visual creativity rather than control or planning tasks, which are central to our focus.
> >
> > Additionally, Genie is trained on a massive dataset comprising 55 million videos totaling 20,000 hours of data, which is significantly beyond the scale of the datasets we use for our work.  The training of Genie involves substantial computational resources. The largest model (2.7B parameters) requires 16+ days of training on 256 TPUv3, and even the smallest variant (41M parameters) takes 3 days on 64 TPUv2. These compute requirements are beyond what is feasible in our current setup. Furthermore, its low frame rate (~1 FPS) makes it unsuitable for efficient planning tasks in control environments.
> >
> > Furthermore, Genie is neither open-sourced nor does it provide access to its trained weights or datasets, as noted in its broader impact statement. This makes it impossible to adapt its model directly for controlled comparison. Genie's pretrained models are not open-sourced, which further limits our ability to perform a direct comparison, as we cannot use the model for evaluation in our environments.
> >
> > Given these factors, a direct comparison with Genie is not feasible. However, we appreciate the opportunity to highlight the differences in scale and purpose between Genie and our work.
> >
> > # Conclusion
> >
> > We hope our responses have sufficiently addressed your questions, and we thank you again for your thoughtful and constructive feedback. In light of this, we kindly ask the reviewer to consider increasing their support for the paper. We are more than happy to discuss any further questions the reviewer may have or provide additional details!
> >
> > # References
> >
> > [1] Genie: Generative Interactive Environments

---

> > > ### Comment · Reviewer_Cbfu · 2024-11-26
> > > **Response to authors**
> > >
> > > Thank you for addressing my concerns and including additional experiments on scaling laws and ablations in the appendix!
> > >
> > > I am inclined towards accepting this paper, but I still don't fully understand the argument against not benchmarking Genie for the control tasks. Eventhough the Genie paper talks about generating interactive environments I don't think there is a fundamental limitation to the method proposed in that paper that won't allow us to run it on same dataset and tasks used in this paper. Regarding the scale of data used in original paper - Yes I agree that is a valid concern but you can just test how well it does in the setting you have and see if scaling dataset size helps. Regarding open-source implementation of the method I believe there are public implementations available like this one https://github.com/1x-technologies/1xgpt/tree/main/genie. Given the review period and effort involved in running this experiment I don't expect authors to run any new experiments. I just wanted to highlight the fact that I don't see a fundamental reason why it won't be possible to benchmark genie style methods (not the pretrained model but one trained from scratch) on the tasks authors are working on.

---

> > > > ### Author Response · Authors · 2024-11-26
> > > >
> > > > Thank you for reviewing our rebuttal and acknowledging that we have addressed your concerns. We also appreciate the resources on the open-sourced implementation of Genie. We are currently working on setting up Genie in our framework and will provide planning evaluations, even though the original Genie paper did not include these evaluations.
> > > >
> > > > That said, we note that Genie is not directly applicable as a baseline in our setting for several reasons:
> > > > - A significant focus of Genie is on the inference of latent actions, particularly in datasets that lack action labels. In contrast, our problem assumes access to ground truth actions.
> > > > - The primary results in the Genie paper emphasize generating imagined videos in out-of-domain environments. This is distinct from our focus, which is on planning performance within the same environment, leveraging world models for reasoning and planning.
> > > >
> > > > Despite these differences, we are implementing Genie and will compare its performance with DINO-WM in our evaluations.
> > > >
> > > > Thank you for your patience, and we will do our best to address your concerns thoroughly.

---

> > > > > ### Author Response · Authors · 2024-12-02
> > > > >
> > > > > Thank you for your patience. We are writing to provide updated experimental results on Genie, following your reference to its public implementation (https://github.com/1x-technologies/1xgpt/tree/main/genie). We would like to note that the provided implementation of Genie does not include a latent action model or action conditioning. To address this, we implemented action conditioning and evaluated two versions of the model:
> > > > > - **Genie**: The original version of the provided Genie implementation.
> > > > > - **Genie-Action**: Genie conditioned on ground truth actions encoded by a trainable MLP.
> > > > > We provide the LPIPS scores on the predicted frames of each model:
> > > > > | Model      | LPIPS $\downarrow$  |
> > > > > |------------|--------|
> > > > > | Genie      | 0.107  |
> > > > > | Genie-Action   | 0.043  |
> > > > > | Ours       | 0.007  |
> > > > >
> > > > > We have also included visualizations of Genie and Genie-Action predictions [here](https://imgur.com/a/4DH2HHh), as well as at the bottom of our anonymous [project website](https://anon-dino-wm.github.io/).
> > > > >
> > > > > From these results, it is evident that Genie performs worse in prediction quality and future state estimation compared to DINO-WM, even with ground truth action conditioning. We hope this addresses your concerns about this baseline. Please let us know if you have any further questions!

---

### Official Review · Reviewer_iuyC · 2024-11-03

**Soundness:** 3
**Presentation:** 3
**Contribution:** 2
**Rating:** 6
**Confidence:** 4

**Summary:**

The authors present a world model to facilitate solving control problems in the physical world. The world model learns the dynamic of the environments in the latent space based on pre-trained DINOv2 that extracts both spatial-aware and object-centric latent representation from 2D images. The approach enables gradient-based trajectory optimization at inference time to deal with various control problems (maze navigation, robot manipulation, etc.). Moreover, it achieves so by only using offline trajectory data during model learning.

**Strengths:**

+ The proposed world model learns to generalize to different variants of several tasks (maze navigation, robot manipulation, etc.) by using only the offline trajectory data. This makes it a potential candidate for training robot foundation models with large quantities of offline data.
+ The trajectory optimization-based test-time control enabled by the proposed world model achieves good results in a range of navigation and manipulation tasks.
+ The comparison of DINOv2 over others (e.g., R3M) as the pre-trained backbone for the world model gives us insights into how good these visual foundation models are for facilitating solving control problems.

**Weaknesses:**

- As the authors have pointed out, latent world models are not a new thing. While the proposed framework might be unique in terms of a reconstruction-free reward-free latent-space world model, I do feel the overall novelty is a little bit limited here.
- While the authors have shown results on a range of control tasks, IMO they are more or less 2D tasks (e.g., in the particular manipulation the particles mostly move on a plane). I am curious how will the approach perform for more complicated manipulation or control tasks (e.g., Humanoid Run from DMControl and Move Chair from ManiSkill2). These tasks might require a much more accurate level of physical reasoning capabilities (objects have momentum and move in a more complex way) which can help us understand better the limit of DINOv2.

**Questions:**

- What is the inference speed of the framework? This show how well they can potentially deal with tasks that heavily require reactive controls.
- Missing references to some recent work that uses Transformers [1] or Diffusion Models [2] to learn to perform latent/visual planning for control problems from offline data.

[1] Chain-of-Thought Predictive Control

[2] Diffusion World Model: Future Modeling Beyond Step-by-Step Rollout for Offline Reinforcement Learning

---

> ### Author Response · Authors · 2024-11-21
>
> We appreciate the reviewer for their insightful feedback, especially for highlighting the potential for DINO-WM “for training robot foundation models with large quantities of offline data.”
>
> We address the issues raised in the review below. We have also updated the manuscript based on the reviewer’s feedback.
> # Overall Novelty
> > "While the proposed framework might be unique in terms of a reconstruction-free reward-free latent-space world model, I do feel the overall novelty is a little bit limited here."
>
> We appreciate the reviewer for acknowledging the novelty of the “reconstruction-free reward-free latent-space world model.” However, as outlined in our global response, we wish to emphasize that the innovation of our work extends beyond these design elements. Specifically, the core novelty and focus lie in addressing the feasibility of constructing **general-purpose world models** that solely capture environment dynamics, enabling reasoning and planning for arbitrary states. This approach targets the long-term goal of creating **task-agnostic world models** capable of generalizing to unseen dynamics and goals. This remains a largely uncharted area in existing research, underscoring the unique contributions of our work.
> # Complexity of Existing benchmarks and Results for Additional Benchmarks
> > “ IMO they are more or less 2D tasks ”
>
> We thank the reviewer for raising this point. We understand that the deformable tasks may appear as a 2D task as particles mostly move in a 2D plane. However, we want to clarify that these tasks are inherently 3D. The observations capture the full 3D space, as illustrated in the visualizations in Figure 4. Furthermore, for robot execution, the system must perform low-level control actions within the simulator, even though a high-level action wrapper is utilized. To provide additional clarity, we have included unskipped rollout visualizations of the dataset on our anonymous website linked in the manuscript.
>
> > “I am curious how will the approach perform for more complicated manipulation or control tasks (e.g., Humanoid Run from DMControl and Move Chair from ManiSkill2)”
>
> While we acknowledge that existing benchmarks feature tasks with lower-level action spaces and greater complexity, tasks like DMControl’s *Humanoid Run* or ManiSkill2’s *Move Chair* demand more than achieving a single target state. These tasks require maintaining continuous motions, such as “walking” or “moving without erratic behavior,” making them unsuitable for offline planning and reasoning approaches. Nevertheless, we conducted an additional experiment on DMControl’s *Reacher-Hard* task, and further enhanced its difficulty by extending it to a multi-goal version. We compared planning performance between DINO-WM and DreamerV3, our strongest baseline, with complete results and visualizations now included in Appendix A.4.
>
> Additionally, we performed further evaluations of DINO-WM on LIBERO [1], a benchmark designed for lifelong robot learning and imitation learning. Here, we demonstrated that DINO-WM, trained on multi-task expert trajectories, effectively distinguishes between expert and non-expert trajectories. This capability positions DINO-WM as a promising approach for integration with multi-task or goal-conditioned policies, enabling the world model to facilitate predictions with high-level goal specifications.
>
> > “These tasks might require a much more accurate level of physical reasoning capabilities (objects have momentum and move in a more complex way) which can help us understand better the limit of DINOv2.”
>
> As the reviewer’s concern is based on the model’s ability to perform accurate physical reasoning for objects with momentum and complex movements, we further trained an unconditional world model on the CLEVRER[2] dataset. This dataset features scenarios with objects of various shapes, sizes, and colors interacting in physically realistic ways, including collisions, rolling, and sliding dynamics. We demonstrated the model’s capability by showcasing predicted future video rollouts, illustrating its understanding of intricate object interactions and dynamics. Videos of these predicted rollouts have been included on our website.
>
> We now address the reviewer’s questions below.

---

> > ### Author Response · Authors · 2024-11-21
> >
> > # Inference Speed
> > > "What is the inference speed of the framework? This show how well they can potentially deal with tasks that heavily require reactive controls."
> >
> > We completely agree that this is an important piece of information for understanding DINO-WM’s performance and applications. We have included the inference time for predicting a batch of future latent state with DINO-WM, the time for planning, as well as how they compare to the inference speed for simulators with expensive computations like the deformable environment suite. This has been included in Appendix A.8.
> > # Missing Citations
> > We thank the reviewer for the references, and have included these citations in the manuscript.
> > # Conclusion
> > We hope the discussion above has answered the reviewer's questions. We ask that, if all their concerns are met, they consider increasing their support for the paper. We thank the reviewer again for their valuable and constructive feedback!
> >
> > # References
> >
> > [1] Bo Liu, Yifeng Zhu, Chongkai Gao, Yihao Feng, Qiang Liu, Yuke Zhu, Peter Stone. LIBERO: Benchmarking Knowledge Transfer for Lifelong Robot Learning
> >
> > [2] Kexin Yi, Chuang Gan, Yunzhu Li, Pushmeet Kohli, Jiajun Wu, Antonio Torralba, Joshua B. Tenenbaum. CLEVRER: CoLlision Events for Video REpresentation and Reasoning

---

> ### Comment · Reviewer_iuyC · 2024-11-26
> **Thanks for the author response**
>
> The author response has addressed my concern. I will leave it up to the AC to decide whether the overall novelty is adequate for the acceptance, though.

---

> > ### Author Response · Authors · 2024-11-26
> >
> > Thank you for taking the time to review our rebuttal and for acknowledging that we have addressed your concerns. We greatly appreciate your thoughtful feedback and evaluation of our work.
> >
> > As noted in our response, to the best of our knowledge, our work is the first to design and evaluate world models for their ability to capture general environment dynamics without being entangled with specific task or goal information. We demonstrate that this capability enables test-time planning for arbitrary, unseen visual targets and varying environment configurations. This addresses an important gap in understanding how embodied agents can reason and infer based on their learned understanding of the world, independent of predefined tasks. We hope this contribution and its novelty is seen as significant in advancing the field.
> >
> > If there are any further clarifications or details we can provide to better highlight our contributions, we would be more than happy to address them.  If you feel our responses and the additional clarifications strengthen the paper, we kindly ask you to consider raising your score.
> >
> > Thank you again for your valuable time and insights!

---

### Official Review · Reviewer_UASa · 2024-11-05

**Soundness:** 3
**Presentation:** 2
**Contribution:** 2
**Rating:** 5
**Confidence:** 4

**Summary:**

This paper introduces a visual dynamics learning approach called DINO-WM, leveraging pre-trained DINOv2 latent features for predictive modeling. DINO-WM incorporates a visual transformer that processes DINO features and action sequences to forecast future observations in an auto-regressive manner, trained through teacher forcing. The model demonstrates zero-shot planning capabilities when integrated with MPC using CEM. The study evaluates DINO-WM across a diverse set of tasks—PointMaze, PushT, Rope, and Granular Manipulation—and achieves
better performance compared to state-of-the-art model-based reinforcement learning approaches.

---

post rebuttal: The authors addressed several concerns. Though I am still worried about the novelty but I will not argue for a rejection as the contribution of the paper is clear. So I increased my score.

**Strengths:**

- Effectively leverages a pre-trained vision model to enhance world model learning.
- Conducts comprehensive evaluations across multiple tasks, validating the model's effectiveness.

**Weaknesses:**

- The paper's writing needs to be improved. The citations should be clearly separated from the main text using brackets when needed.
- Although DINO-WM uses DINOv2's latent features, the approach of learning a visual dynamics model in the latent space is not groundbreaking. Pretraining latent features for world model learning or learning latent world models through rewards are both exploited in previous literature. Leveraging existing vision models is not novel to me.
- In one comparison, the authors evaluate TD-MPC2 without rewards, which might not provide a fair assessment. Since reward signals are central to TD-MPC2’s optimization, omitting them potentially undermines its performance.
- The work does not include results on established image-based RL benchmarks, such as DM-Control or Minecraft, which are widely used to validate model-based approaches like TD-MPC2 and Dreamer. Including these benchmarks would strengthen claims of general applicability​.
- The model’s lack of Q-learning hinders its capacity to handle tasks requiring long-term planning. In RL, Q-learning or similar approaches are often necessary for efficiently solving complex, high-horizon tasks. This limitation may affect DINO-WM’s scalability to challenging environments.​

**Questions:**

- I wonder what causes the low performance of TDMPC, is the reward fully ignored or the reward depends on the latent representation that it just learned?

---

> ### Author Response · Authors · 2024-11-21
>
> Thank you for your constructive feedback. We are delighted that you found our work to have effectively leveraged a pre-trained vision model to enhance world model training, and that our experiments across multiple environments and tasks are comprehensive and showing the model’s effectiveness. We appreciate your insights and will address your questions and concerns below.
>
> # Writing Improvement
> > “The paper's writing needs to be improved. The citations should be clearly separated from the main text using brackets when needed.”
>
> Thank you for your suggestion. We have carefully revised the manuscript to improve the overall writing quality and ensure that citations are clearly separated from the main text using brackets where appropriate.
>
> # Novelty of DINO-WM
>
> > “Although DINO-WM uses DINOv2's latent features, the approach of learning a visual dynamics model in the latent space is not groundbreaking. Pretraining latent features for world model learning or learning latent world models through rewards are both exploited in previous literature. Leveraging existing vision models is not novel to me.”
>
> Thank you for your insight. We agree that learning world models in the latent space itself is not novel. However, while there has been a large body of research using latent world models, they always assume access to downstream tasks and reward information. Therefore, it remains unknown whether it’s possible to build a general-purpose action-conditioned world model, and if they would enable generalization to arbitrary goals and novel environment layouts for control tasks. Our aim is to provide insights into this open question, using a pre-trained vision model as a means to facilitate this exploration, rather than proposing the use of pre-trained vision models as a novelty on its own.
>
> # Evaluations on Established Image-Based RL Benchmarks
>
> > “The work does not include results on established image-based RL benchmarks, such as DM-Control or Minecraft, … Including these benchmarks would strengthen claims of general applicability​.”
>
> We thank the reviewer for pointing this out. While we agree that these benchmarks are widely used for evaluating RL methods, we would like to clarify that our primary objective differs from theirs. Specifically, DINO-WM is designed to operate in an **offline, reward-free setting** where the task is not known during training, and the goal is to generalize to arbitrary environment dynamics and goals at test time.
>
> However, we agree that including some of these benchmarks would provide more insights of general applicability. Therefore, we ran additional experiments on the Reacher-Hard task of DMControl, as well as the *libero_goal* task suite on LIBERO[1], a manipulation benchmark for robotics. We have included these results in Appendix A.4.
>
> # Lack of Q-Learning
>
> > “The model’s lack of Q-learning hinders its capacity to handle tasks requiring long-term planning. In RL, Q-learning or similar approaches are often necessary for efficiently solving complex, high-horizon tasks. This limitation may affect DINO-WM’s scalability to challenging environments.​”
>
> Thank you for the constructive feedback on long-term planning. We agree that handling long-horizon tasks is a significant challenge, particularly in the context of embodied AI. While Q-learning plays a crucial role in model-based RL methods where access to rewards are assumed, it cannot be directly applied in our setting where there is no reward information, and goals are only revealed at test time. We appreciate your suggestion and believe that incorporating techniques such as hierarchical planning could be a promising direction to enhance DINO-WM's ability to address long-term planning challenges in such reward-free and task-agnostic settings.
>
> We now answer the reviewer’s questions.

---

> > ### Author Response · Authors · 2024-11-21
> >
> > # Performance of TDMPC2
> >
> > > “I wonder what causes the low performance of TDMPC, is the reward fully ignored or the reward depends on the latent representation that it just learned?”
> >
> > Great question! We note that for all the datasets we use to train the world models, they don’t have any reward information as the data is supposed to only show environment dynamics, without any notion of “task” or “goal”. Therefore, in order to compare with TDMPC2, we use null rewards for TDMPC2. Since TDMPC2’s encoder is learned with the objective of reconstructing rewards, in our setting without rewards, there’s no sufficient signals to learn a good encoder and hence world model, which explains the poor performance for TDMPC2.
> > # Conclusion
> >
> > We hope our clarifications, along with the additional experiments we included, have addressed your concerns and emphasized the strengths of our work. If our responses have resolved your concerns, we would greatly appreciate it if you could consider adjusting your score. We are happy to continue the discussion and address any further questions you may have!
> >
> > # References
> > [1] Bo Liu, Yifeng Zhu, Chongkai Gao, Yihao Feng, Qiang Liu, Yuke Zhu, Peter Stone. LIBERO: Benchmarking Knowledge Transfer for Lifelong Robot Learning

---

> > > ### Author Response · Authors · 2024-11-26
> > >
> > > Dear reviewer, we sincerely appreciate the time and effort you have dedicated to reviewing our work. As we near the end of the rebuttal period, we welcome any further comments or questions. If we have successfully addressed all your concerns, we kindly ask if you would consider increasing your score. Otherwise, we are more than happy to provide additional clarifications or experiments!

---

> > > > ### Author Response · Authors · 2024-12-02
> > > >
> > > > Dear reviewer, thank you again for taking the time to review our work, and we appreciate the opportunity to improve our paper based on your feedback. As we near the end of the discussion period, we kindly ask if there are any remaining concerns or points of clarification that we can address.
> > > > We value your input and are happy to address any further questions. Thank you again for your time and consideration.

---

### Author Response · Authors · 2024-11-21
**Global Response**

We thank the reviewers for their time and for finding that our work DINO-WM "effectively leverages a pre-trained vision model for world modeling," (**UASa**) has potential for "training robot foundation models," (**iuyC**), is "simple, scalable, and effective," (**Cbfu**) and "compares favorably to task-specific world models." (**fU9L**). There are, however, several significant concerns, especially around comparisons to task-specific RL settings. In this general comment, we would like to address clarifications of our setting and additional experiments that should assuage the reviewers’ concerns.
# Clarification: Offline and Reward-Free Task Solving
Several reviewers (**UASa**, **iuyC**, & **fU9L**) suggested evaluations on established RL benchmarks like DMControl. However, we want to clarify that **DINO-WM is not an task-specific online RL algorithm**. In fact, the entire motivation behind DINO-WM is that embodied agents need to be able to solve a variety of tasks instead of a single one. Most RL benchmarks such as DMControl typically assume:
- Access to rewards throughout training
- Knowledge of the task or goals during training
- Access to simulators for online interactions

In contrast, DINO-WM operates in a purely offline and reward-free setting. It is trained without tasks, goals, or rewards, and the evaluation focuses on the generalization abilities of the learned world model to arbitrary goals and environment layouts at test time. Nevertheless, when trying to find appropriate DMControl tasks, we found that the Reacher task can be adapted to a multi-goal setting, and have added results to the paper.
# Clarification: Novelty of This Work
Reviewers **UASa** & **iuyC** highlighted that prior work also leverages “latent world models” or “use pre-trained vision models for world models.” We want to clarify that our work **does not claim these choices as novel contributions**. Instead, the novelty of this work lies in addressing whether it's possible to build general-purpose world models that exclusively capture environment dynamics and enable reasoning and planning to solve arbitrary states. This long-term objective of task-agnostic world models, capable of generalizing to unseen dynamics and goals, remains largely unexplored in prior work and represents the true potential of world models.
# Additional Experiments
## Evaluations on DMControl and LIBERO (**UASa**, **iuyC**, & **fU9L**)
To address reviewers’ suggestions on evaluations on established RL benchmarks, we include additional experiments of the Reacher task in DMControl in the manuscript in Appendix A.4. We observe a 43% performance increase than DreamerV3, our most competitive baseline.

We also evaluate on LIBERO[1], a task suite featuring offline imitation learning, and show that learned world models can be effectively used to distinguish expert vs. non-expert trajectories in Appendix A.4.
## Evaluations on Diffusion-based Baselines (AVDC) (**Cbfu**)
While the original AVDC can’t be applied for planning as they only provide a sequence of observations instead of executable actions, we further introduce an action-conditioned AVDC variant, and report its planning performance in Appendix A.7.
## Scaling Laws of DINO-WM
Reviewer **Cbfu** suggests experiments scaling with the amount of training data. We agree that this could provide more insights on the scalability and data efficiency of DINO-WM, and we show results in Appendix A.5.1.
## DINO-WM with and without Causal Mask
Reviewer **fU9L** suggested ablation experiments on the predictor's added causal mask. We thank the reviewer for this constructive feedback, and included results in Appendix A.5.2.
## Backprop Decoder Loss to Predictor
We follow Reviewer **fU9L**’s suggestion on a variant of the model where reconstruction loss from the decoder is back propagated to the predictor, and updated the results in Appendix A.5.3.
# Additional Results
## Inference Speed
Reviewer **iuyC** raised this important point for understanding the applicability of DINO-WM. We include the inference speed and planning time for DINO-WM in Appendix A.8.
## Physical Reasoning on Complex Tasks
To show DINO-WM’s physical reasoning capabilities mentioned by Reviewer **iuyC**, we provide two new visualizations in our anon website:
- Full rollouts of deformable tasks involving 7-DoF robotic arms to highlight tasks beyond simple 2D manipulation.
- Unconditioned predictions on CLEVRER[2] featuring multiple 3D objects sliding and colliding. These demonstrate the model's ability to reason about physical interactions accurately.
## Training Details
We include additional training hyperparameters and model architectures in Appendix A.9, following Reviewer **Cbfu**’s feedback.

We welcome further discussions and are happy to provide additional results or clarifications as needed.
# References
[1] LIBERO: Benchmarking Knowledge Transfer for Lifelong Robot Learning

[2] CLEVRER: CoLlision Events for Video REpresentation and Reasoning

---

### Author Response · Authors · 2024-11-26

We sincerely thank all reviewers for their thoughtful and constructive feedback. In response, we have made several updates and added experiments to address the points raised:
- Evaluations on the Reacher task of DMControl, showing a 43% performance improvement of DINO-WM compared to DreamerV3, our most competitive baseline. (**UASa**, **iuyC** & **fU9L**)
- Evaluations on the LIBERO[1] benchmark, demonstrating DINO-WM’s ability to identify expert vs. non-expert trajectories. (**UASa**, **iuyC** & **fU9L**)
- Quantitative planning evaluations of an action-conditioned AVDC variant as a diffusion-based baseline. (**Cbfu**)
- Ablation experiments with varying dataset sizes, showing DINO-WM's scaling laws. (**Cbfu**)
- Ablation experiments demonstrating the effectiveness of the causal mask introduced in DINO-WM. (**fU9L**)
- Ablation comparing DINO-WM with and without decoder loss, highlighting the benefits of decoupling world model training from reconstruction objectives. (**fU9L**)
- Inference and planning speed evaluations showing DINO-WM’s wall-clock time efficiency over simulators with complex dynamics. (**iuyC**)
- Evaluations on the CLEVRER[2] dataset, showing DINO-WM’s physical reasoning capabilities for objects with complex dynamics and behaviors like colliding, rotating, and sliding. (**iuyC**)
- Included additional training details and model hyperparameters in the appendix of the updated manuscript. (**Cbfu**)
- Added missing references, improved citation formats, and refined writing per reviewer suggestions. (**UASa**, **iuyC**, **Cbfu** & **fU9L**)
- Addressed reviewers’ questions regarding baselines, clarifications of DINO-WM, and comparisons with recent world model style work like Genie[3]. (**UASa**, **iuyC**, **Cbfu** & **fU9L**)

We are very grateful for your time and your help in improving our work, and we believe that we have addressed all reviewers’ feedback and questions. As the rebuttal period closes to its end, we kindly ask reviewers to review our responses and the updated manuscript. We are happy to answer any additional questions you might have!

[1] Bo Liu, Yifeng Zhu, Chongkai Gao, Yihao Feng, Qiang Liu, Yuke Zhu, Peter Stone. LIBERO: Benchmarking Knowledge Transfer for Lifelong Robot Learning

[2] Kexin Yi, Chuang Gan, Yunzhu Li, Pushmeet Kohli, Jiajun Wu, Antonio Torralba, Joshua B. Tenenbaum. CLEVRER: CoLlision Events for Video REpresentation and Reasoning

[3] Genie: Generative Interactive Environments

---

### Meta-Review · Area_Chair_rBq9 · 2024-12-19

**Metareview:**

The paper presents DINO-WM, a method for learning world models that leverage pre-trained DINOv2 visual features to enable zero-shot planning. The core idea is to predict future states in the latent space of DINOv2's patch features, trained purely from offline trajectories without requiring rewards or online interaction.

Strengths: The paper demonstrates empirically that pre-trained visual features can serve as a useful representation space for world models. The evaluations are thorough, showing strong results on maze navigation, pushing, and particle manipulation tasks compared to baselines like DreamerV3 and TD-MPC2.

However, the paper has a fundamental disconnect between its claimed contribution and actual analysis. The authors identify a key challenge in offline world models: without sufficient data coverage, these methods typically require auxiliary learning objectives (like reconstruction or rewards) to learn a good latent planning space (lines 53-83). The paper proposes that using DINOv2's pre-trained features, which contain spatial and object-centric priors, can address this challenge by enabling effective planning without such auxiliary objectives.

Yet this core claim lacks rigorous justification in two critical ways. First, the paper does not establish that auxiliary objectives actually help compensate for limited data coverage - this baseline connection is necessary to motivate their solution. Second, while empirical results show DINOv2 features are effective, the paper fails to analyze why these features provide a good planning space without auxiliary learning. Without understanding what properties of DINOv2 (geometric, semantic, or physical) enable this capability, it's unclear if the method truly addresses the stated coverage challenge or simply works well for unrelated reasons. A rigorous comparison with other pre-trained representations, analyzing their relevant properties for dynamics modeling, would be necessary to properly justify the proposed solution.

Moreover, the paper falls short in benchmarking against established robotics tasks (like DMControl or RLBench) where random exploration is insufficient. While the authors added some comparisons during rebuttal, the results remain limited to simpler scenarios. The strong assumptions about data collection and the lack of a clear path to scaling to more complex tasks suggest this approach.

Given these limitations, particularly the lack of rigorous analysis explaining why DINOv2 features enable better world modeling, I recommend rejecting this paper.

**Additional Comments On Reviewer Discussion:**

Initial Reviews:
- UASa questioned the novelty of learning world models in latent space and using pre-trained visual features
- iuyC raised concerns about the paper's focus on 2D tasks and requested inference speed information
- Cbfu noted inconsistency in describing offline data requirements and requested comparisons with diffusion-based methods
- fU9L had concerns about insufficient benchmarks and the assumption that random exploration provides sufficient coverage

Author Responses:
- DMControl Reacher task showing 43% improvement over DreamerV3
- LIBERO benchmark evaluations for trajectory assessment
- Quantitative comparisons with action-conditioned AVDC variant
- Scaling analysis with varying dataset sizes
- Ablation studies on causal masking

Reviewer Reactions after Rebuttal:
- UASa maintained concerns about novelty but increased their score, acknowledging the paper's clear contribution
- iuyC requested Genie comparisons but was satisfied with the authors' explanation of why direct comparison wasn't feasible
- Cbfu was satisfied with the additional experiments and clarifications
- fU9L raised their score after seeing the additional benchmarks but maintained some reservations about the offline learning setting

---

### Decision · Program_Chairs · 2025-01-22

Reject